# Homology-guided identification of a conserved motif linking the antiviral functions of IFITM3 to its oligomeric state

**Kazi Rahman[1†], Charles A Coomer[1,2†], Saliha Majdoul[1], Selena Y Ding[1], Sergi Padilla-Parra[2,3,4], Alex A Compton[1]***

[1]HIV Dynamics and Replication Program, Center for Cancer Research, National Cancer Institute, Frederick, United States; [2]Cellular Imaging Group, Wellcome Trust Centre for Human Genetics, University of Oxford, Oxford, United Kingdom; [3]Department of Infectious Diseases, King's College London, Faculty of Life Sciences & Medicine, London, United Kingdom; [4]Randall Division of Cell and Molecular Biophysics, King's College London, London, United Kingdom

**Abstract** The interferon-inducible transmembrane (IFITM) proteins belong to the Dispanin/CD225 family and inhibit diverse virus infections. IFITM3 reduces membrane fusion between cells and virions through a poorly characterized mechanism. Mutation of proline-rich transmembrane protein 2 (PRRT2), a regulator of neurotransmitter release, at glycine-305 was previously linked to paroxysmal neurological disorders in humans. Here, we show that glycine-305 and the homologous site in IFITM3, glycine-95, drive protein oligomerization from within a GxxxG motif. Mutation of glycine-95 (and to a lesser extent, glycine-91) disrupted IFITM3 oligomerization and reduced its antiviral activity against Influenza A virus. An oligomerization-defective variant was used to reveal that IFITM3 promotes membrane rigidity in a glycine-95-dependent and amphipathic helix-dependent manner. Furthermore, a compound which counteracts virus inhibition by IFITM3, Amphotericin B, prevented the IFITM3-mediated rigidification of membranes. Overall, these data suggest that IFITM3 oligomers inhibit virus-cell fusion by promoting membrane rigidity.

**\*For correspondence:**
alex.compton@nih.gov

[†]These authors contributed equally to this work

**Competing interests:** The authors declare that no competing interests exist.

## Introduction

The intrinsic protection of cells from virus infection represents an early and essential aspect of antiviral innate immunity. Cytokines, including interferons, signal the presence of invading viruses and induce an 'antiviral state' via the expression of hundreds of antiviral genes (*Yan and Chen, 2012*; *Bieniasz, 2004*). This arsenal of antiviral proteins converges on many steps of the virus life cycle in order to collectively inhibit infection of cells and prevent virus spread. In addition, certain 'front-line' antiviral proteins impose a constant barrier to infection because they are expressed constitutively and are further upregulated by interferons. The interferon-induced transmembrane (IFITM) proteins are the earliest acting restriction factors known, inhibiting the entry of diverse viruses into cells by restricting fusion pore formation during virus-cell membrane fusion (*Shi et al., 2017*; *Chesarino et al., 2014a*; *Perreira et al., 2013*; *Bailey et al., 2014*). Among the growing list of viruses shown to be inhibited by IFITM proteins in cell culture and in vivo are orthomyxoviruses, flaviviruses, filoviruses, alphaviruses, and coronaviruses (*Shi et al., 2017*). IFITM3 is a potent inhibitor of Influenza A virus (IAV) infection in cell culture and in vivo, and consequently, it is the most studied member of the IFITM family (*Bailey et al., 2012*; *Everitt et al., 2012*; *Allen et al., 2017*). While the precise mechanism by which IFITM3 reduces virus-cell fusion remains unresolved, evidence suggests that it does so by altering the properties of lipid membranes.

Two models have been proposed to explain how IFITM3 inhibits virus fusion. In the first, IFITM3 plays an indirect role by interacting with VAMP-associated protein A (VAPA) and inhibiting lipid transport between the endoplasmic reticulum and endosomes, resulting in an accumulation of endosomal cholesterol (*Amini-Bavil-Olyaee et al., 2013*). High cholesterol content may inhibit the fusion of virus-containing vesicles with the limiting membrane of the late endosome, restricting virus release into the host cell cytoplasm. While inhibition of cholesterol trafficking has been shown to inhibit virus entry (*Poh et al., 2012*), this model of IFITM3 function has been subsequently supported by one study (*Kühnl et al., 2018*) but challenged by several others (*Appourchaux et al., 2019*; *Lin et al., 2013*; *Desai et al., 2014*; *Wrensch et al., 2014*). The second and currently favored model posits that IFITM3 directly inhibits fusion by locally reducing membrane fluidity (i.e. increasing membrane rigidity or order) (*Lin et al., 2013*; *Li et al., 2013*) and by inducing positive membrane curvature (*Li et al., 2013*). Membrane order and curvature influence one another (*Vanni et al., 2014*; *Rangamani et al., 2014*; *Golani et al., 2019*) and, together, they regulate numerous membrane fusion processes (*Martens and McMahon, 2008*). Consistent with the notion that IFITM3 induces local membrane order and curvature to disfavor virus-cell fusion, IFITM3 contains a juxtamembrane amphipathic helix which is essential for antiviral activity (*Chesarino et al., 2017*). Amphipathic helices have been identified in many eukaryotic, prokaryotic, and viral proteins and are well known for their ability to bind and bend membranes (*Drin and Antonny, 2010*; *Giménez-Andrés et al., 2018*). Another piece of evidence that supports the membrane deformation model of fusion inhibition is that IFITM3 incorporated into enveloped viruses also impairs virion fusion with target cells (*Tartour et al., 2014*; *Yu et al., 2015*; *Tartour et al., 2017*; *Ahi et al., 2020*; *Sharma et al., 2019*; *Compton et al., 2016*; *Suddala et al., 2019*; *Compton et al., 2014*). Despite this progress, a complete mechanistic view of IFITM3 is lacking because studies describing the impact of IFITM3 on membranes have not included mutants lacking antiviral function.

It was previously reported that IFITM3 forms clusters on virus-containing vesicles (*Kummer et al., 2019*) and that IFITM3 oligomerization promotes restriction of IAV (*John et al., 2013*). However, the determinants initially purported to mediate oligomerization (phenylalanines at residues 75 and 78 of the CD225 domain [*John et al., 2013*]) were later shown to be unnecessary when oligomerization was measured using a FRET-based approach in living cells (*Winkler et al., 2019*). Therefore, the oligomerization of IFITM3 appears to be influenced by unknown determinants and its importance to antiviral function is not established.

In the current study, we set out to identify a loss-of-function mutation in IFITM3 suitable for mechanistic studies by using a homology-guided approach. The *IFITM* genes (*IFITM1*, *IFITM2*, *IFITM3*, *IFITM5*, and *IFITM10* in humans) are members of an extended gene family known as the Dispanin/CD225 family (hereafter referred to as CD225 proteins) (*Sällman Almén et al., 2012*; *Zhang et al., 2012*). Members of this group are characterized by the presence of a CD225 domain, but the functions of most remain unknown. However, one member is the subject of numerous studies because it is linked to neurological disorders. Mutations in proline-rich transmembrane protein 2 (PRRT2) result in conditions of involuntary movement, such as paroxysmal kinesigenic dyskinesia, benign familial infantile seizures, and episodic ataxia (*Gardiner et al., 2015*; *Valtorta et al., 2016*). PRRT2 is a neuron-specific protein that is localized to pre-synaptic terminals and which inhibits synaptic vesicle fusion (*Meschia, 2018*; *Mo et al., 2019*; *Liu et al., 2016*). Molecular studies of disease-associated missense mutations in *PRRT2* (G305W/R) indicate that it causes loss-of-function, leading to unchecked neurotransmitter release (*Valente et al., 2016*; *Coleman et al., 2018*; *Gardiner et al., 2012*; *van Vliet et al., 2012*; *Liu et al., 2012*). Interestingly, the homologous residue in human IFITM3 is also subject to rare allelic variation in humans (G95W/R) and this mutation results in partial loss of activity against IAV infection (*John et al., 2013*). However, the reason why this site is essential for the respective functions of PRRT2 and IFITM3 was unknown.

Here, we demonstrate that glycine-95 of human IFITM3 resides within a GxxxG motif that is highly conserved among vertebrate IFITM3 orthologs as well as PRRT2. Mutation of glycine-91 or glycine-95 rendered IFITM3 less active against IAV (in target cells) and HIV-1 (in virus-producing cells). We found that the GxxxG motif mediates IFITM3 oligomerization in living cells, with glycine-95 playing a dominant role. An IFITM3 mutant (G95L) exhibiting loss of antiviral function was deficient for oligomerization, indicating that IFITM3 oligomerization and virus restriction are functionally associated. We leveraged this loss-of-function mutant to identify mechanistic correlates of antiviral function which are associated with IFITM3 oligomerization. We found that IFITM3 increased membrane order,

as previously suggested, whereas IFITM3 encoding G95L or mutations within its amphipathic helix failed to do so. In an effort to further probe the importance of membrane order in the antiviral mechanism, we demonstrate that Amphotericin B (Ampho B) decreases the stiffness of IFITM3-containing membranes and rescues virus infection. These data indicate that promotion of membrane order by IFITM3 oligomers is required for its antiviral activity. Furthermore, we reveal that oligomerization is a shared requirement for the distinct anti-fusion functions performed by homologs IFITM3 and PRRT2.

## Results

### Homology-guided identification of a putative oligomerization motif within CD225 domains

While it has been suggested that IFITM3 may adopt multiple topologies (*Li et al., 2013*; *Yount et al., 2012*), experimental evidence indicates that IFITM3 is a type II transmembrane protein characterized by the presence of a cytoplasmic-facing amino terminus, a CD225 domain consisting of a hydrophobic intramembrane (IM) domain, a cytoplasmic intracellular loop (CIL), and a hydrophobic transmembrane (TM) domain, and a very short carboxy terminus facing the vesicle lumen or extracellular space (*Bailey et al., 2013*; *Ling et al., 2016*). PRRT2 is thought to adopt a similar topology in membranes (*Rossi et al., 2016*). We used Protter, an interactive application that maps annotated and predicted protein sequence features onto the transmembrane topology of proteins (*Omasits et al., 2014*) to visualize IFITM3 and PRRT2. The two proteins exhibit a similar predicted topology consisting of dual hydrophobic domains and a CIL, but the amino terminus is considerably longer in PRRT2 (*Figure 1A and B*). Given the association of G305W with loss-of-function of PRRT2, we wondered whether mutation of the homologous residue in IFITM3 (glycine-95) would compromise its respective functions as well. Upon comparing topologies and the protein alignment, we noticed that these glycines form part of a GxxxG motif in the CIL of IFITM3 and PRRT2, and this motif is intact in several other IFITM and CD225 proteins (*Figure 1C*). The GxxxG motif, also known generally as a (small)xxx(small) motif, is frequently associated with dimerization of membrane proteins (*Teese and Langosch, 2015*; *Overton et al., 2003*). Most often shown to mediate pairing of hydrophobic transmembrane helices within a bilayer, the motif has also been described to drive oligomerization from cytoplasmic loops or linkers (*Lu et al., 2014*). The GxxxG motif is conserved in *IFITM3* of vertebrates, indicating that it may play an important functional role (*Figure 1D*).

### $^{91}$GxxxG$^{95}$ is important for restriction of virus entry by IFITM3

We generated FLAG-tagged IFITM3 mutants in which glycine-91 and glycine-95 were changed to leucine (G91L and G95L), following an example set by characterization of the GxxxG motif in the human folate transporter (*Wilson et al., 2015*). We also produced a G95W mutant because a rare single-nucleotide polymorphism in *IFITM3* known as rs779445843 gives rise to missense mutations (G95W/R) in human populations (*Cunningham et al., 2019*; *Sherry et al., 2001*). Furthermore, G95W in IFITM3 is analogous to the disease-associated polymorphism in PRRT2 (G305W) which results in loss of function (*Coleman et al., 2018*).

To begin the functional characterization of IFITM3 harboring mutations within its GxxxG motif, we assessed steady-state protein levels following transient and stable transfection into HEK293T cells. Transiently expressed IFITM3 mutants reached similar levels as wild-type (WT) as determined by flow cytometry (*Figure 1—figure supplement 1A–B*). However, western blot analysis revealed a significant expression defect for the G91L mutant, but not for the others (*Figure 1—figure supplement 1C–D*). In stably transfected cells, all IFITM3 constructs resulted in protein expression levels that exceeded those observed following transient transfection, and the only mutant exhibiting significantly reduced levels was G95W (*Figure 2A and B*).

Next, we used confocal immunofluorescence microscopy to address how modification of the GxxxG motif of IFITM3 influences its subcellular localization. In cells stably or transiently expressing IFITM3 constructs, we assessed the extent to which IFITM3 colocalizes with EEA1-GFP, a marker for early endosomes, and CD63, a marker for late endosomes/multivesicular bodies (*Figure 2—figure supplement 1A–B*). As shown previously (*Shi et al., 2017*; *Huang et al., 2011*; *Feeley et al., 2011*), IFITM3 WT was detected in early endosomes, late endosomes, and at the plasma membrane.

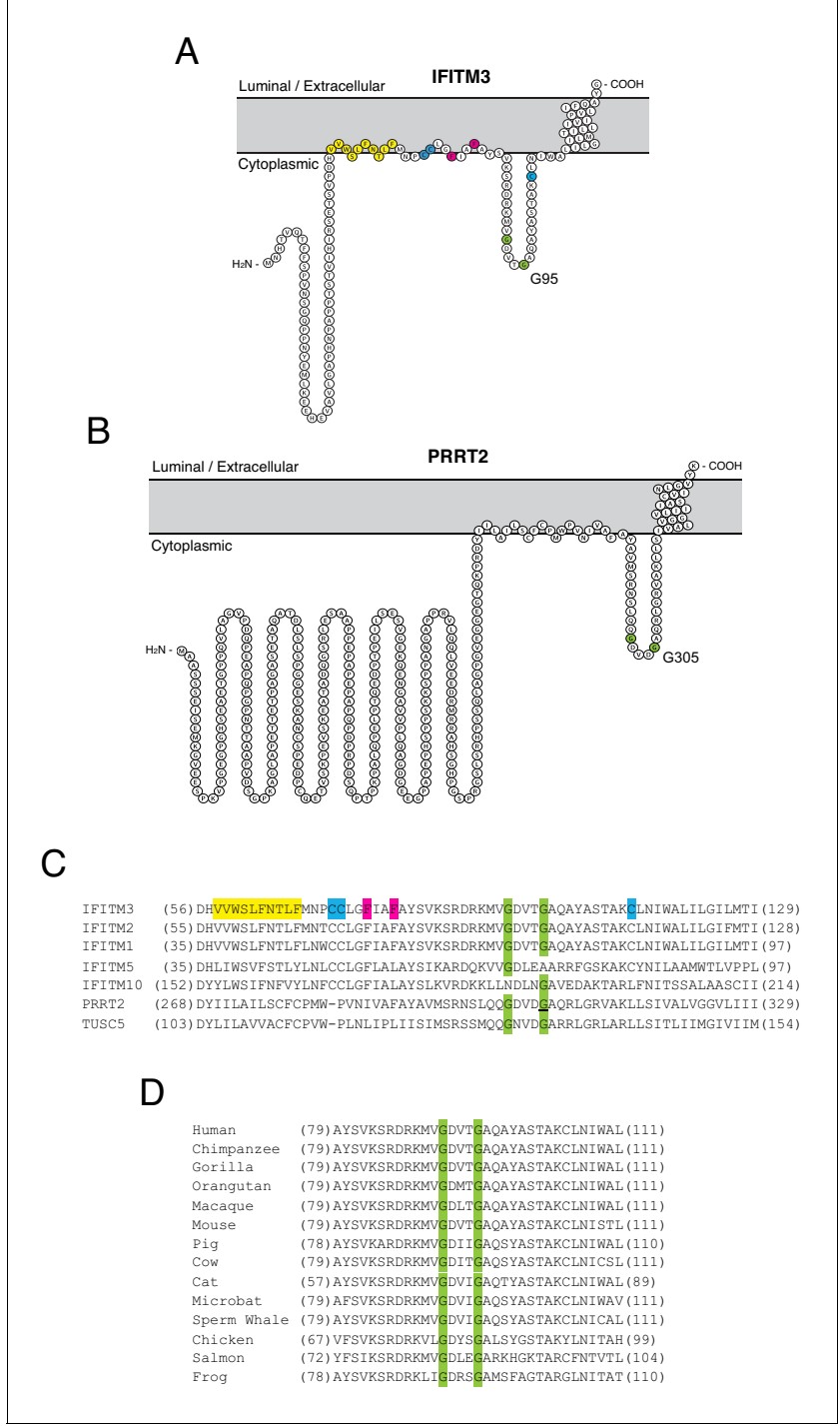

**Figure 1.** Homology-guided identification of a putative oligomerization motif within CD225 domains. (**A**) Schematic representation of the membrane topology of IFITM3 made with Protter. Residues corresponding to the amphipathic helix (yellow), palmitoylated cysteines (blue), phenylalanines purported to regulate oligomerization (red), and the glycines of the GxxxG motif (green) are indicated. (**B**) Schematic representation of the membrane topology of PRRT2 made with Protter. Residues corresponding to the glycines of the GxxxG motif (green) are indicated. (**C**) A partial amino acid alignment of CD225 domains from IFITM proteins, PRRT2, and TUSC5. Color codes are included as in (**A** and **B**). The position of the polymorphic glycine in PRRT2 associated with neurological disease (G305W) is underlined. (**D**) A partial amino acid alignment of IFITM3 orthologs in vertebrates. Conserved glycines in the GxxxG motif (green) are indicated.

The online version of this article includes the following figure supplement(s) for figure 1:

**Figure supplement 1.** Quantitative measurement of IFITM3 construct expression following transient transfection and flow cytometric analysis of virus-cell fusion.

Specifically, approximately 30% of IFITM3 WT was associated with early endosomes and 30% was associated with late endosomes. The G91L, G95L, and G95W mutations did not obviously alter colocalization between IFITM3 and endosomes in either stably or transiently transfected cells (*Figure 2—figure supplement 1A–B*). We noticed that stable expression of all IFITM3 constructs resulted in greater colocalization with CD63+ late endosomes. We also noticed that the G91L mutation resulted in an increased colocalization between IFITM3 and both early endosomes and late endosomes under transient conditions, but this difference was not statistically significant. Therefore, it is unlikely that mutations in the GxxxG motif impact the subcellular localization of IFITM3.

HEK293T cell lines stably expressing FLAG-tagged IFITM3 WT and mutants were then challenged with IAV to assess antiviral function. IFITM3 WT strongly protected cells from infection, but G91L resulted in a partial loss of virus restriction while G95L and G95W resulted in a more substantial loss of restriction (*Figure 2C and D*). The extent of restriction did not correlate with relative levels of IFITM3 protein in stable cell lines (*Figure 2B*), demonstrating that differential antiviral activity is not due to differential protein expression. To confirm that IFITM3 targets the virus entry step, we challenged cells with HIV-1 pseudotyped with VSV glycoprotein (VSV-G) in an assay for virus-cell fusion (*Figure 1—figure supplement 1E* and *Figure 2E*). Inhibition of HIV-VSV-G by the IFITM3 constructs resembled inhibition of IAV, in that G95L and G95W resulted in a substantial loss of restriction. G91L, however, only slightly impacted virus restriction and its activity was not significantly different from WT. Therefore, a rational approach identified glycine-95 to be a major determinant for the broad inhibition of virus entry by IFITM3. Since the G95L and G95W mutations were functionally redundant, we did not pursue the characterization of G95W further.

## $^{91}$GxxxG$^{95}$ is important for restriction of HIV-1 virion infectivity by IFITM3

We and others have previously demonstrated that, in addition to preventing virus entry into naive target cells, IFITM3 performs another antiviral function in virus-producing (infected) cells by incorporating into virions, reducing viral glycoprotein abundance and function, and reducing the fusogenic potential of virions (*Tartour et al., 2014*; *Yu et al., 2015*; *Tartour et al., 2017*; *Ahi et al., 2020*; *Sharma et al., 2019*; *Compton et al., 2014*). Therefore, we tested the impact of mutations in the GxxxG motif on restriction of HIV-1 virion infectivity. IFITM3 WT reduced the infectivity of HIV-1, while G95L resulted in a partial loss of activity. Strikingly, G91L completely abrogated this antiviral function (*Figure 3A and B*).

In order to identify correlates of antiviral function against HIV-1, we assessed viral Envelope (Env) levels expressed in virus-producing cells and levels incorporated into virions. The transient expression of IFITM3 WT in virus-producing cells was accompanied by a partial loss of Env gp120 and gp41 in those cells (*Figure 3C*) and in virions (*Figure 3D*) as previously reported (*Yu et al., 2015*; *Ahi et al., 2020*; *Wang et al., 2017*). This effect was measured in replicate experiments using quantitative immunoblotting (*Figure 3E and F*). In contrast, transient expression of the G91L or G95L mutants did not result in reduced Env levels (*Figure 3C–F*). Since the G91L and G95L mutations differentially impact the restriction of HIV-1 virion infectivity by IFITM3, it is therefore unlikely that Env quantity in virions fully accounts for this restriction. However, we observed that the G91L and G95L mutations strongly impaired the ability of IFITM3 itself to incorporate into virions (*Figure 3G*), and the extent of virion incorporation correlated with the measured impact on HIV-1 infectivity (*Figure 3B*). Together, our results demonstrate that the dual antiviral functions performed by IFITM3 (early-stage inhibition of virus entry and late-stage inhibition of virion infectivity) are critically regulated by the GxxxG motif. Interestingly, the former function depends primarily on glycine-95, while the latter function depends more on glycine-91.

## $^{91}$GxxxG$^{95}$ regulates oligomerization of IFITM3 in living cells

Based on their position within a conserved GxxxG motif, we used Förster resonance energy transfer (FRET) to assess the roles played by glycine-91 and glycine-95 in the oligomerization of IFITM3. We constructed IFITM3 fused with yellow fluorescent protein (YFP) or mCherry at the amino terminus to create FRET pairs and to perform fluorescence lifetime imaging microscopy (FLIM). The measurement of fluorescence lifetimes allows for measurements that are independent of fluorophore expression level or diffusion rate (*Day, 2014*). In this framework, excitation of YFP (donor) results in energy

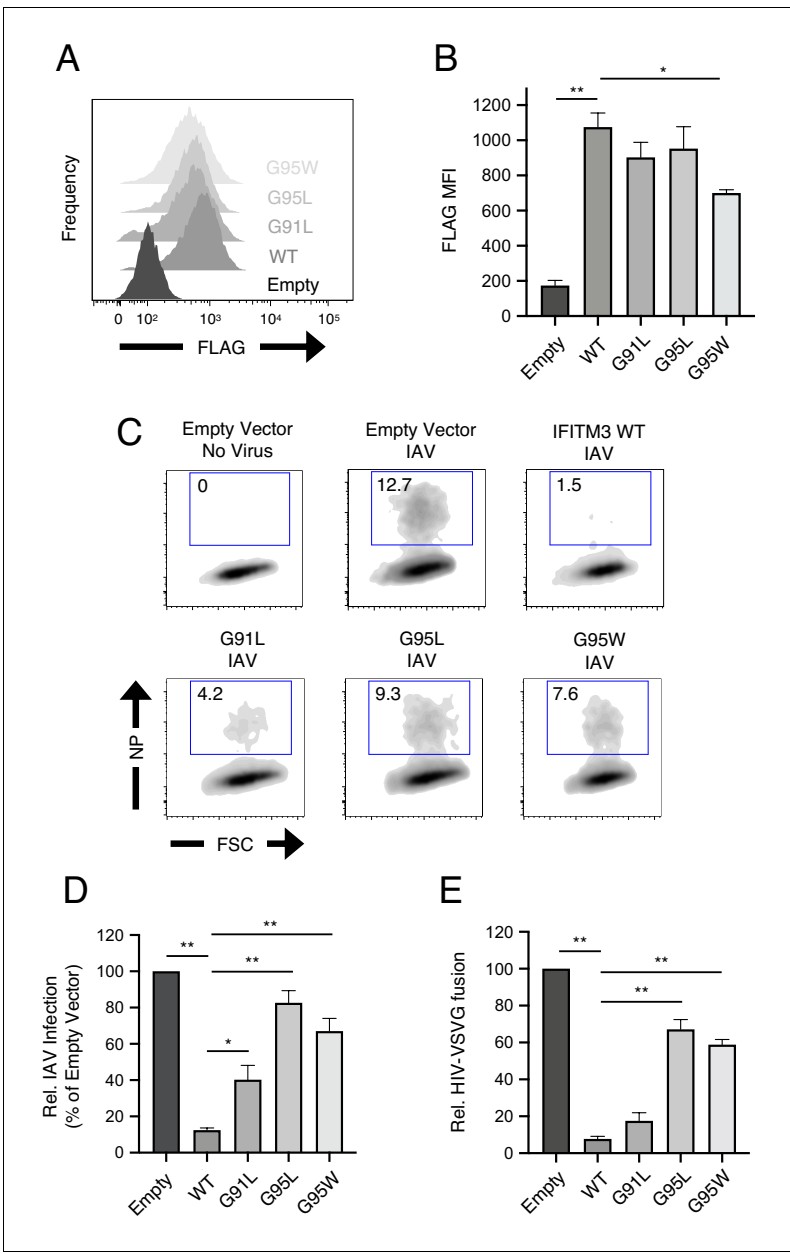

**Figure 2.** $^{91}$GxxxG$^{95}$ is important for restriction of virus entry by IFITM3. (**A**) HEK293T cells stably transfected with empty pQCXIP, IFITM3 WT-FLAG, or the indicated mutants were fixed, stained with anti-FLAG antibody and assessed by flow cytometry. FLAG levels are displayed as histograms. (**B**) Mean fluorescence intensity measurements of FLAG staining in (**A**) are shown for three independent experiments. (**C**) HEK293T cells stably transfected with empty pQCXIP, IFITM3 WT-FLAG, or the indicated mutants were challenged with IAV PR8 strain (MOI of 0.1), fixed at 18 hr post-infection, stained with an anti-nucleoprotein antibody, and assessed by flow cytometry. (**D**) Mean infection results representing 5–8 independent experiments are normalized to empty vector (set to 100%). (**E**) HEK293T cells stably transfected with empty pQCXIP, IFITM3 WT-FLAG, or the indicated mutants were challenged with replication-incompetent HIV-1 incorporating BlaM-Vpr and pseudotyped with VSV glycoprotein. Virus-cell fusion was assessed at 2.5 hr post-virus addition using the beta lactamase assay and flow cytometry. Results represent the mean of three independent experiments and are normalized to empty vector (set to 100%). Error bars indicate standard error. Statistical analysis was performed using one-way ANOVA. *, $p < 0.05$; **, $p < 0.001$. MFI, mean fluorescence intensity. Rel., relative.

The online version of this article includes the following figure supplement(s) for figure 2:

**Figure supplement 1.** The subcellular localization of IFITM3 WT and mutants as measured by confocal immunofluorescence microscopy.

transfer to mCherry (acceptor) when a molecular interaction brings the pair into close proximity. Co-transfection of mCherry and IFITM3 WT-YFP served to establish background FRET values. By comparison, IFITM3 WT-YFP and IFITM3 WT-mCherry co-transfection resulted in high levels of detectable FRET (*Figure 4A and B*). Relative to WT, introduction of the G91L mutation partially inhibited FRET, while the G95L mutation had a much stronger effect, reducing FRET to near background levels (*Figure 4A and B*). These data suggest that both glycine-91 and glycine-95 contribute to IFITM3 oligomerization, but that glycine-95 is the major determinant. FRET was also measured for heterologous pairs (combining IFITM3 WT-YFP and G95L-mCherry, and vice versa) and the results were indicative of an intermediate degree of oligomerization (*Figure 4B*). To complement our FRET analysis, we also analyzed fluorescence lifetimes of the donor (YFP). Co-transfection of IFITM3 WT-YFP and IFITM3 WT-mCherry resulted in significant decreases in YFP lifetimes (averaging about 400 picoseconds), consistent with IFITM3 oligomerization (*Figure 4A and C*). Meanwhile, co-transfection of G91L-YFP and G91L-mCherry resulted in partial decreases in YFP lifetimes (averaging about 225 picoseconds), while we observed only minor decreases in YFP lifetimes (averaging about 70 picoseconds) when G95L-YFP and G95L-mCherry were examined. Within this experimental framework, we confirmed that the F75A/F78A mutations did not affect IFITM3 oligomerization, nor did mutations within the amphipathic helix (S61A/N64A/T65A) (*Figure 4—figure supplement 1*). Since our experiments revealed that glycine-95 is the major determinant for IFITM3 oligomerization, we focused on the G95L mutant for further mechanistic characterization.

To ensure that conclusions drawn from this imaging approach are relevant to IFITM3-mediated antiviral function, we tested the ability for our fluorescently-tagged IFITM3 constructs to inhibit IAV. The antiviral potency of IFITM3 WT tagged with mCherry or YFP at the amino terminus was decreased by 20% compared to IFITM3 WT-FLAG (*Figure 4—figure supplement 2A–B*). Nonetheless, IFITM3 G95L tagged with mCherry or YFP exhibited no antiviral activity whatsoever, validating the utility of our fluorescently-tagged IFITM3 constructs for mechanistic studies. During the preparation of this manuscript, Suddala et al. reported the construction and use of an IFITM3 fusion protein encoding a fluorescent protein placed internally, rather than terminally, after residue 40 of IFITM3 (*Suddala et al., 2019*). Since this fusion protein was purported to exhibit minimal loss to antiviral function, we decided to produce additional fusion proteins following a similar strategy. We introduced mCherry or YFP into IFITM3 after residue 40 as described in Suddala et al. and assessed these constructs for antiviral activity and oligomerization potential. We found that the internal placement of mCherry or YFP resulted in a greater loss of antiviral activity compared to the amino terminal placement of mCherry or YFP. Specifically, the constructs encoding mCherry or YFP internally lost more than 40% of their activity compared to a 20% loss exhibited by constructs encoding mCherry or YFP at the amino terminus (*Figure 4—figure supplement 2A–B*). For thoroughness, we tested the IFITM3 WT constructs encoding internal mCherry and YFP for FRET competence and found that oligomerization was apparent (*Figure 4—figure supplement 2C*). Furthermore, introduction of the G95L mutation into these constructs resulted in decreased oligomerization (*Figure 4—figure supplement 2C*). Therefore, these data demonstrate that our FRET-based approach to studying IFITM3 oligomerization is amenable to the placement of fluorophores at the amino terminus or after residue 40 of IFITM3. However, placement of YFP or mCherry at the amino terminus allows for better preservation of antiviral function.

## Glycine-95 regulates oligomerization of IFITM3 in denaturing and native conditions

In parallel to our studies of IFITM3 oligomerization in single, living cells, we assayed the ability of IFITM3 pairs tagged with FLAG or myc to co-immunoprecipitate from bulk cell lysates. HEK293T were co-transfected with IFITM3-FLAG and IFITM3-myc followed by FLAG immunoprecipitation, SDS-PAGE, and quantitative immunoblotting. We found that IFITM3 WT-myc readily pulled down with IFITM3 WT-FLAG, while pull down of G95L-myc with G95L-FLAG was diminished by approximately 50% (*Figure 5A and B*). To address heteromultimerization between IFITM3 WT and IFITM3 G95L, we paired IFITM3 WT-FLAG with G95L-myc and the results were indicative of an intermediate degree of oligomerization (*Figure 5—figure supplement 1A–B*). Therefore, membrane-extracted IFITM3 forms oligomers, but the G95L mutation reduces oligomerization. We then performed blue native PAGE and immunoblotting to assess the oligomeric state of IFITM3 under non-denaturing conditions. Two populations of IFITM3 oligomers, exhibiting sizes of approximately 300 and 480

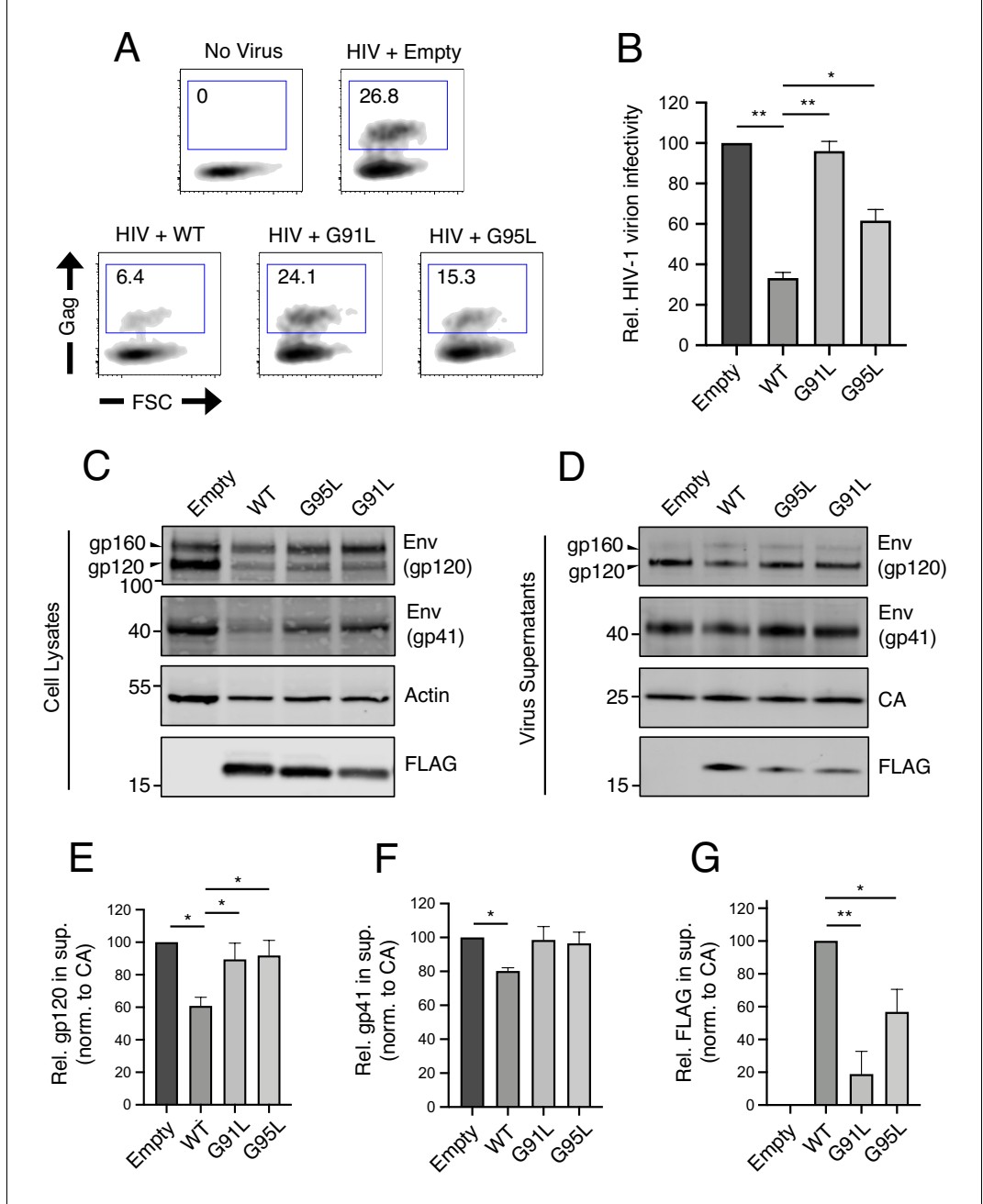

**Figure 3.** $^{91}$GxxxG$^{95}$ is important for restriction of HIV-1 virion infectivity by IFITM3. (**A**) HEK293T were co-transfected with HIV-1 molecular clone pNL4.3 and empty pQCXIP, IFITM3 WT-FLAG, or the indicated mutant. Virus-containing supernatants were harvested at 48 hr post-transfection and subjected to ultracentrifugation over sucrose pellets. Virion content was quantified by p24 CA ELISA and 50 ng p24 equivalent was added to TZM.bl cells for infectivity measurements. TZM.bl were fixed at 48 hr post-infection, stained with anti-CA antibody, and infection was assessed by flow cytometry. (**B**) Mean infection results of TZM.bl using virus derived from three independent transfections of HEK293T are shown and normalized to empty vector (set to 100%). (**C**) Whole cell lysates and (**D**) virus-containing supernatants were collected from HEK293T co-transfected with the HIV-1 molecular clone pNL4.3 and empty pQCXIP, IFITM3 WT-FLAG, or the indicated mutant at 48 hr post-transfection. Virus-containing supernatants were ultracentrifuged through sucrose cushions. Both lysates and concentrated, purified virus-containing supernatants (50 ng p24 equivalent) were subjected to SDS-PAGE and Western blot analysis. Immunoblotting was performed with anti-gp120, anti-gp41, anti-CA, anti-actin, and anti-FLAG. (**E**) Virion-associated levels of gp120 Env were quantified by measuring fluorescence of DyLight-conjugated secondary antibody and were normalized to levels of CA in three independent experiments. (**F**) Virion-associated levels of gp41 Env and (**G**) IFITM3-FLAG were quantified similarly. For anti-Env immunoblotting, the amount of gp120 or gp41 in virions was presented relative to empty vector (set to 100%). For anti-FLAG immunoblotting, the amount of IFITM3 WT in virions was set to 100%. Error bars indicate standard error. Statistical analysis was performed using one-way ANOVA. *, $p<0.05$; **, $p<0.001$. Rel., relative. Norm., normalized.

kilodaltons, were readily observed for IFITM3 WT-FLAG and, to a lesser extent, IFITM3 G95L-FLAG. The largest (480 kilodaltons) was reduced by approximately 50% for IFITM3 G95L (*Figure 5C and D*). A complete scan of the blue native PAGE result in *Figure 5C* is shown in *Figure 5—figure supplement 1C*. We did not detect a population of IFITM3 dimers (expected at approximately 30 kilodaltons), which may reflect the specific conditions under which blue native PAGE was performed here. The fact that IFITM3 G95L exhibited a reduced potential for higher order oligomer formation using this technique is consistent with our experiments using living cells or denatured bulk lysates. Therefore, glycine-95 is necessary for efficient IFITM3 oligomer formation.

To quantitatively resolve the specific oligomeric state of IFITM3, we performed a Number and Brightness analysis (*Digman et al., 2008*; *Nolan et al., 2017*). This approach is a fluorescence fluctuation microscopy method capable of measuring the apparent average number of molecules and their brightness in each pixel over time, with brightness being proportional to oligomeric state. We restricted our analysis to brightness of IFITM3-mCherry as it trafficked in and out of the plasma membrane over time. On average, IFITM3 WT-mCherry was found to be 2.14 times brighter than mCherry monomers (*Shaner et al., 2004*; *Figure 5E*). In contrast, G95L-mCherry brightness was not significantly different than mCherry monomers (averaging 1.30 times brighter). Furthermore, relative to mCherry monomers, both IFITM3 WT-mCherry and G95L-mCherry formed assemblies that were up to five times brighter, but IFITM3 WT-mCherry demonstrated a greater propensity to form these higher order oligomers (*Figure 5E*). One caveat of this Number and Brightness analysis is that, in contrast to transmembrane proteins, mCherry monomers do not target membranes and are expressed mostly in the cytoplasm (*Teese and Langosch, 2015*). However, the fact that IFITM3 WT exhibits a mean brightness that is roughly twofold greater than IFITM3 G95L supports the notion that the former exists primarily as a dimer and the latter as a monomer. Together, these data suggest that IFITM3 forms dimers and higher order oligomers in membranes in a glycine-95-dependent fashion.

## Membrane order is increased by IFITM3 oligomers and is a correlate of antiviral function

While the precise mechanism by which IFITM3 inhibits virus-cell fusion remains unresolved, a salient phenotype of IFITM protein expression is increased membrane order (reduced membrane fluidity) (*Lin et al., 2013*; *Li et al., 2013*; *John et al., 2013*). Therefore, we leveraged our loss-of-function mutant, G95L, to directly test whether membrane order is functionally associated with inhibition of virus entry by IFITM3. Previous reports of membrane order enhancement by IFITM family members involved the use of a cell-permeable dye known as Laurdan (*Zhang et al., 2006*). Here, we assessed membrane order using Laurdan and a recently described sensor known as fluorescent lipid tension reporter (FliptR) (*Colom et al., 2018*; *Goujon et al., 2019*; *Coomer et al., 2020*). FliptR is a planarizable push-pull probe that incorporates efficiently into artificial and living cell membranes and whose fluorescence parameters change upon alterations in local lipid packing (order). Specifically, FliptR responds to increasing membrane order at the plasma membrane and at endomembranes by planarization, leading to longer fluorescence lifetimes detected by FLIM (*Colom et al., 2018*; *Dal Molin et al., 2015*). Using FliptR in HEK293T stably expressing IFITM3-FLAG, we found that IFITM3 WT expression led to significantly increased membrane order (*Figure 6A*). In fact, the IFITM3-induced enhancement of membrane order was similar to that achieved by addition of soluble cholesterol, while cholesterol depletion with methyl-beta-cyclodextrin resulted in profound decreases in membrane order (*Figure 6—figure supplement 1A–B*). In contrast to WT, cells expressing IFITM3 G95L did not exhibit increased membrane order (*Figure 6A*). Similar results were obtained using Laurdan, in that IFITM3 WT expression resulted in significantly increased membrane order while G95L did not (*Figure 6—figure supplement 1C–D*). These data indicate that membrane order enhancement tracks with a functionally competent form of IFITM3 but not with a loss-of-function mutant. To further probe the functional importance of membrane order in the antiviral mechanism of IFITM3, we performed experiments in the presence of Ampho B. This antimycotic polyene compound was previously reported to overcome the antiviral activity of IFITM3, rendering cells stably expressing IFITM3 fully permissive to IAV (*Lin et al., 2013*). However, it was unknown how Ampho B counteracts the effects of IFITM3. When we added Ampho B to cells expressing IFITM3 WT, we no longer observed increased membrane order (*Figure 6A*). Furthermore, in identically treated cells, Ampho B prevented restriction by IFITM3 of HIV pseudotyped with hemagglutinin (HA) from IAV (*Figure 6B*).

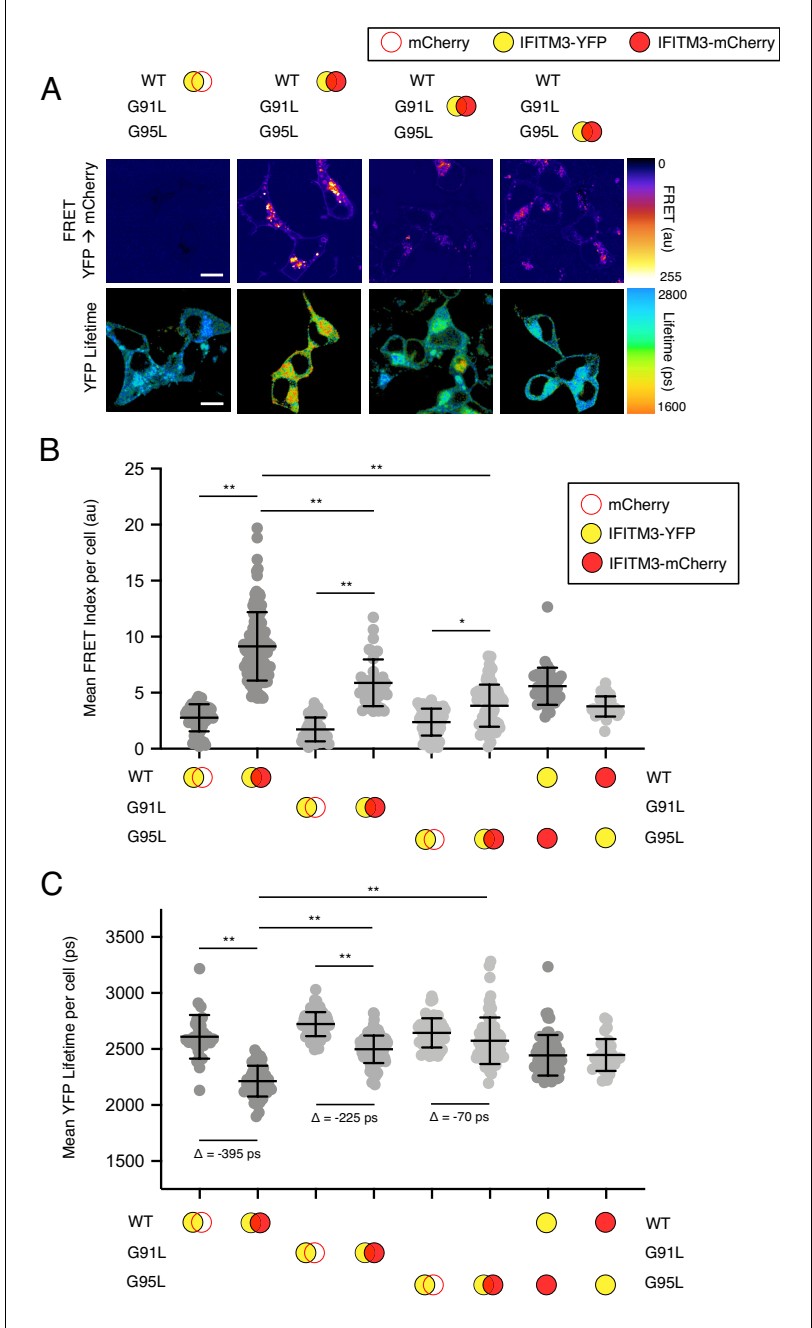

**Figure 4.** $^{91}$GxxxG$^{95}$ regulates oligomerization of IFITM3 in living cells. (**A**) HEK293T were transiently co-transfected with IFITM3-YFP and mCherry or IFITM3-YFP and IFITM3-mCherry. Constructs encoded IFITM3 WT, IFITM3 G91L, or IFITM3 G95L. FRET-FLIM measurements were made, and images of FRET signal and YFP lifetimes are representative of 12–20 captured images per condition. (**B**) Whole-cell FRET analysis was performed on a minimum of 50 cells per condition and the results of three independent experiments were pooled. Dots correspond to individual cells. (**C**) Whole-cell YFP lifetimes were measured on a minimum of 50 cells per condition and the results of three independent experiments were pooled. Dots correspond to individual cells. A mean delta (Δ) value is indicated to represent the drop in YFP lifetime resulting from the pairing of IFITM3-YFP and mCherry versus the pairing of IFITM3-YFP and IFITM3-mCherry. Empty red circles are used to depict mCherry, filled red circles are used to depict IFITM3-mCherry (either WT, G91L, or G95L), and filled yellow circles are used to depict IFITM3-YFP (either WT, G91L, or G95L). Error bars indicate standard deviation. Statistical analysis was performed using one-way ANOVA. *, p<0.05; **, p<0.001. Scale bars, 10 µm. Ps, picoseconds. Au, arbitrary units.

The online version of this article includes the following figure supplement(s) for figure 4:

**Figure supplement 1.** Assessment of additional IFITM3 mutations on oligomerization measured by FRET-FLIM.

**Figure supplement 2.** Assessing the functional impact of fluorescent protein placement within IFITM3 fusion proteins.

These findings show, for the first time, that the capacity for Ampho B to overcome the antiviral activity of IFITM3 is linked to its ability to decrease membrane order. Therefore, the use of Ampho B as an interrogative tool reinforced the role played by membrane order in the antiviral mechanism of IFITM3.

Since we previously reported a critical role for the amphipathic helix of IFITM3 in virus restriction (*Chesarino et al., 2017*), we tested whether the helix was also required for the enhanced membrane order observed in cells expressing IFITM3. Like the G95L mutant, a version of IFITM3 encoding S61A/N64A/T65A mutations in the amphipathic helix lost the ability to increase membrane order (*Figure 6—figure supplement 1E*). Since the S61A/N64A/T65A mutations did not inhibit oligomerization (*Figure 4—figure supplement 1*), our results identify oligomerization and the amphipathic helix as dual requirements for the antiviral functions of IFITM3. Overall, these data strongly suggest that virus restriction by IFITM3 occurs through oligomerization-dependent membrane stiffening induced by the amphipathic helix.

## Disease-associated G305W impairs oligomerization of PRRT2

Since the GxxxG motif is a shared feature of IFITM3 and PRRT2 and that a naturally occurring mutant G305W is predictive of neurological disease, we assessed the oligomerization capacity of WT and mutant PRRT2. As in *Figure 4*, we constructed PRRT2 fused with YFP or mCherry to create FRET pairs. We observed that co-transfection of PRRT2 WT-YFP and PRRT2 WT-mCherry resulted in significant FRET, demonstrating that PRRT2 oligomerizes in living cell membranes (*Figure 7A*). However, FRET was significantly reduced for pairs containing G305W (G305W-YFP and G305W-mCherry) (*Figure 7A*). Furthermore, pairs containing G305W did not exhibit loss in YFP lifetimes relative to their WT counterparts (*Figure 7B*). These data suggest that mutation of glycine-305 in PRRT2 results in loss of protein oligomerization. Therefore, the divergent functions played by homologues IFITM3 and PRRT2 in the regulation of fusion processes are controlled by a common determinant.

## Discussion

The physiological importance of IFITM3 in the control of many virus infections in vivo is becoming increasingly apparent (*Zani and Yount, 2018*; *Kenney et al., 2017*). While it has been proposed that membrane remodeling by IFITM3, at the level of membrane order and curvature, protects host cells from virus invasion, functional proof was lacking. However, two recent developments provided a glimpse of how (and when) IFITM3 inhibits virus-cell membrane fusion. First, the identification of an amphipathic helix located in the IM domain of IFITM3 provided a rational explanation for how membrane stiffening and/or bending may occur (*Chesarino et al., 2017*). Second, IFITM3 has been observed to intercept vesicles carrying inbound virions and to restrict their release into the cytoplasm, while viruses that are insensitive to IFITM3 evade its encounter (*Suddala et al., 2019*; *Spence et al., 2019*). Together with the previous demonstrations that the subcellular localization of IFITM3 determines its specificity and potency (*Compton et al., 2016*; *Jia et al., 2012*; *Chesarino et al., 2014b*), a 'proximity-based' mechanism presents itself in which IFITM3 interacts with and modifies host membranes needed by some viruses to fuse with cells. Importantly, this model accommodates the antiviral effect of IFITM3 inside viral lipid bilayers as well (*Tartour et al., 2014*; *Yu et al., 2015*; *Ahi et al., 2020*; *Suddala et al., 2019*; *Compton et al., 2014*).

Here, we provide evidence that an additional determinant (protein oligomerization) plays a crucial role within this mechanistic framework. Previously, an alanine scanning approach led to the identification of two residues (phenylalanine-75 and phenylalanine-78) that were important for IFITM3 oligomerization in cell lysates (*John et al., 2013*). This study did not functionally test a role for glycine-91 or glycine-95 in the oligomerization of IFITM3 because alanine mutagenesis overlapping these residues apparently led to loss of stable protein expression (*Appourchaux et al., 2019*; *John et al., 2013*). While it has been confirmed that F75A and F78A mutations disrupt antiviral activity (*Suddala et al., 2019*; *Zhao et al., 2018*), IFITM3 oligomerization was not impacted by these mutations when assessed by FRET in living cells (*Winkler et al., 2019*) and we confirm here that F75A and F78A mutations do not interfere with the measurement of IFITM3 oligomerization by FRET. Instead, we find that G91L partially abrogates oligomerization while G95L almost completely disrupts oligomerization. Moreover, our data shows that G95L strongly reduces the antiviral potential of IFITM3 as well as its capacity to increase membrane order (reduce membrane fluidity).

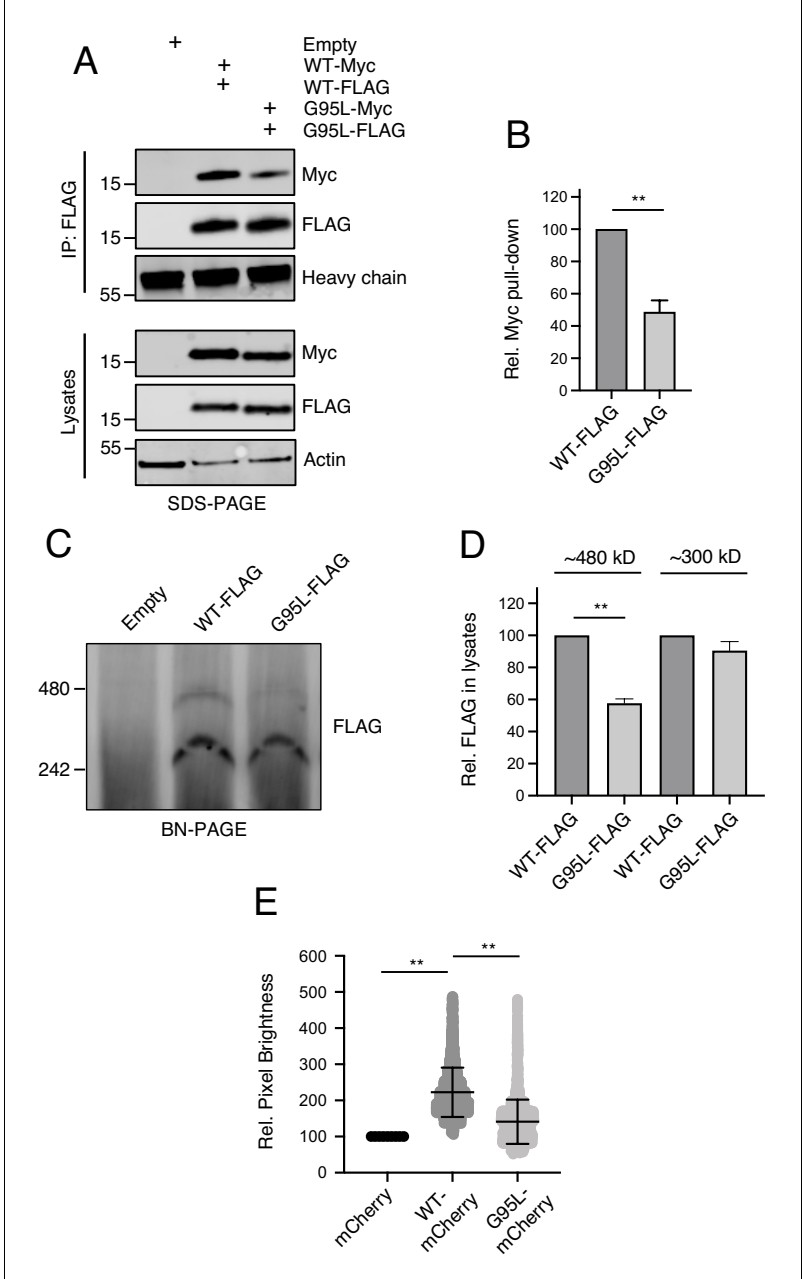

**Figure 5.** Glycine-95 regulates oligomerization of IFITM3 in denaturing and non-denaturing conditions. (**A**) HEK293T were transiently transfected with empty pQCXIP or the following pairs: IFITM3 WT-FLAG and IFITM3 WT-myc or G95L-FLAG and G95L-myc. Whole cell lysates were produced under mildly denaturing conditions and immunoprecipitation (IP) using anti-FLAG antibody was performed. IP fractions and volumes of whole cell lysates were subjected to SDS-PAGE and Western blot analysis. Immunoblotting was performed with anti-FLAG and anti-myc. Heavy chain IgG and Actin served as loading controls in the IP fraction and lysates fraction, respectively. Number and tick marks indicate size (kilodaltons) and position of protein standards in ladder. (**B**) Levels of IFITM3-myc (either WT or G95L) co-immunoprecipitated by anti-FLAG IP were quantified in (**A**) and represented as the mean of three independent experiments. Error bars indicate standard error. (**C**) HEK293T were transiently transfected with empty pQCXIP, IFITM3 WT-FLAG or G95L-FLAG. Cell lysates were produced with 1% digitonin and blue native PAGE was performed, followed by immunoblotting with anti-FLAG. Number and tick marks indicate size (kilodaltons) and position of protein standards in ladder. (**D**) Levels of IFITM3-FLAG (either WT or G95L) corresponding to ~480 kd and ~300 kD were quantified in (**C**) and represented as the mean of three independent experiments. Error bars indicate standard error. (**D**) Number and Brightness analysis was performed on monomeric mCherry and IFITM3-mCherry (either WT or G95L) as described in the Materials and methods. Statistical analysis was performed using student's T test. *, p<0.05; **, p<0.001. Rel., relative.

The online version of this article includes the following figure supplement(s) for figure 5:

**Figure supplement 1.** Blue native PAGE of IFITM3 and assessment of heteromultimerization between IFITM3 WT-FLAG and G95L-myc.

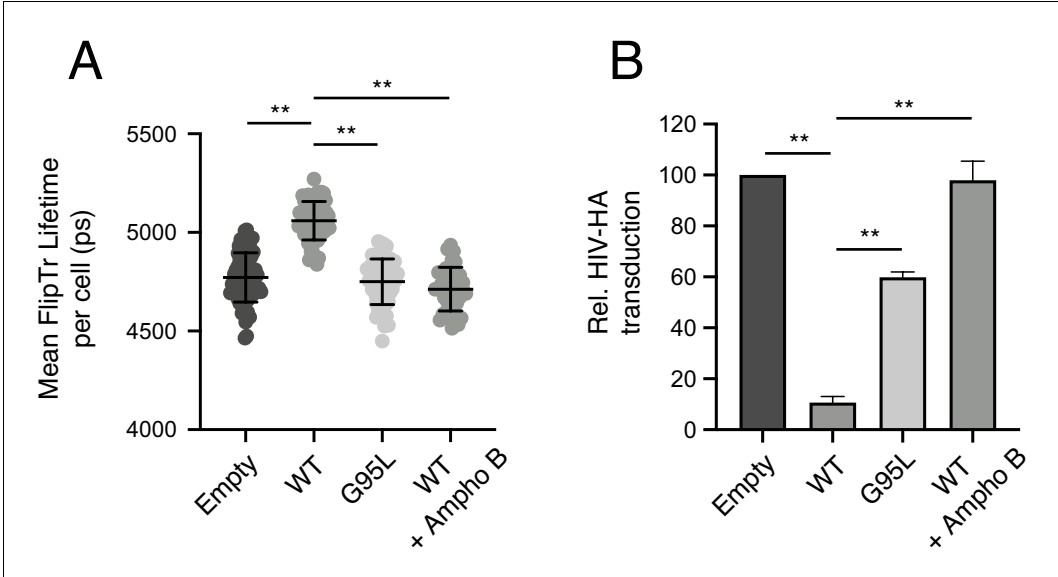

**Figure 6.** Membrane order is increased by IFITM3 oligomers in an Ampho B-sensitive manner. (**A**) HEK293T cells stably transfected with empty pQCXIP, IFITM3 WT-FLAG, or G95L-FLAG were stained with 1 µM FliptR for 5 min and imaged by FLIM. In the condition indicated, 1 µM Amphotericin B was added to cells for 1 hr and washed away prior to addition of FliptR and imaging. The whole-cell mean fluorescence lifetime ($\tau$), in addition to individual component lifetimes (long and short lifetimes, $\tau_1$ and $\tau_2$), was calculated using Symphotime for a minimum of 40 cells per condition and $\tau_1$ from three independent experiments were pooled and plotted. Dots correspond to individual cells. Error bars indicate standard deviation. (**B**) HEK293T cells stably transfected with empty pQCXIP, IFITM3 WT-FLAG, or G95L-FLAG were challenged with HIV pseudotyped with hemagglutinin (HA) from IAV WSN strain at a MOI of 1. In the condition indicated, 1 µM Amphotericin B was added to cells for 1 hr prior to virus addition. Cells were fixed at 48 hr post-infection, and infection was scored by GFP expression using flow cytometry. Statistical analysis was performed using one-way ANOVA. *, $p < 0.05$; **, $p < 0.001$. Ps, picoseconds. Rel., relative.

The online version of this article includes the following figure supplement(s) for figure 6:

**Figure supplement 1.** Assessment of cholesterol addition and cholesterol depletion on membrane order by FliptR and Laurdan.

---

Furthermore, we show that a compound previously found to abrogate the antiviral function of IFITM3, Ampho B (*Lin et al., 2013*; *Suddala et al., 2019*), decreases membrane order in IFITM3-expressing cells. Even though the exact mechanism by which Ampho B impacts mammalian membranes is unclear (*Lin et al., 2013*; *Kamiński, 2014*), these results identify that the membrane stiffening property of IFITM3 is a strict correlate of its antiviral functions in cells and, perhaps, in virions.

We found that glycine-91 and glycine-95 of the GxxxG motif were also important for restriction of HIV-1 virion infectivity by IFITM3. However, the G91L mutation had a stronger impact than G95L on this antiviral activity in virus-producing cells, suggesting that IFITM3 oligomerization is not as crucial for the restriction of HIV-1 infectivity as it is for the restriction of incoming virus in target cells. De novo HIV-1 assembly occurs at the plasma membrane of virus-producing cells, and this is the site where cellular IFITM3 is incorporated into virions (*Compton et al., 2016*). Importantly, we noticed that the majority of the FRET signal occurring between IFITM3-YFP and IFITM3-mCherry was detected in intracellular endosomes and other membrane vesicles, while FRET at the cell surface was relatively low. Therefore, IFITM3 oligomers may be more abundant in endosomal membranes and this agrees with our finding that viruses entering cells via fusion with endosomes (such as IAV) are strongly restricted by IFITM3 oligomers. Since the results presented in this manuscript strongly suggest that IFITM3-mediated antiviral activity occurs through oligomerization-dependent membrane stiffening, assessment of how virion-associated IFITM3 impacts the order of the virion membrane will be helpful in determining whether a distinct antiviral mechanism is at play there.

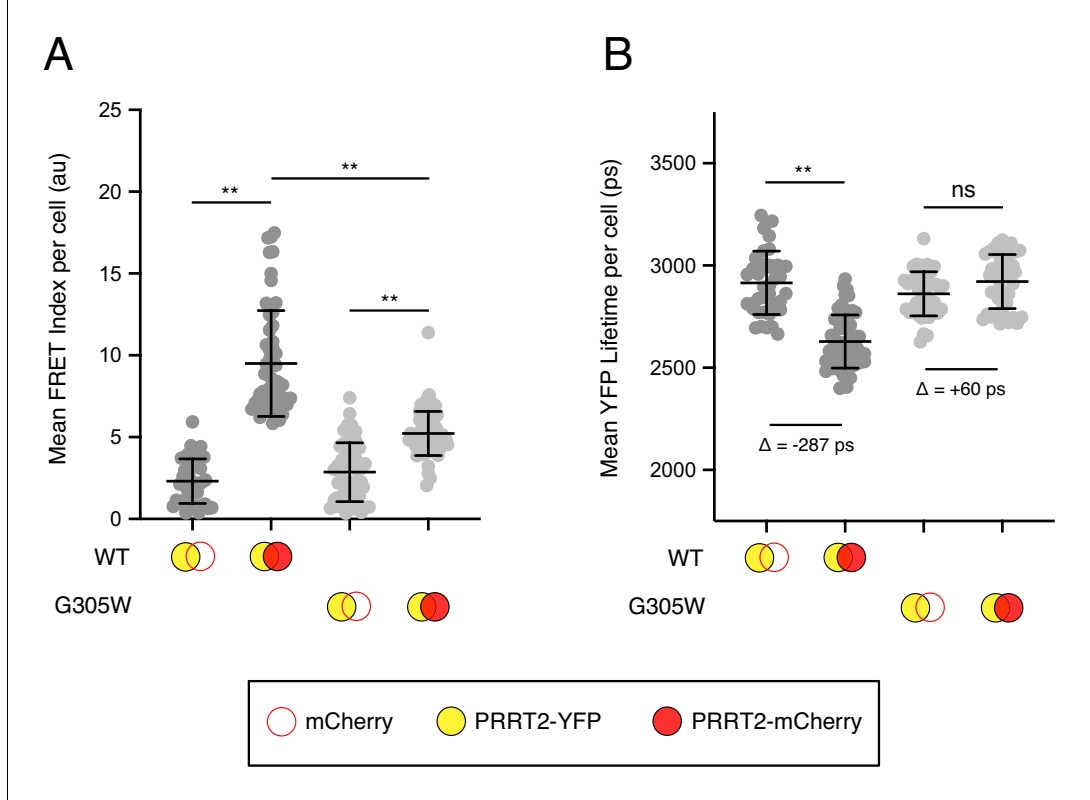

**Figure 7.** Disease-associated G305W disrupts the oligomerization of PRRT2 in living cells. (**A**) HEK293T were transiently co-transfected with PRRT2-YFP and mCherry or PRRT2-YFP and PRRT2-mCherry. Constructs encoded PRRT2 WT or PRRT2 G305W. Whole-cell FRET analysis was performed on a minimum of 50 cells per condition, and the results of three independent experiments were pooled. Dots correspond to individual cells. (**B**) Whole-cell YFP lifetimes were measured on a minimum of 50 cells per condition and the results of three independent experiments were pooled. Dots correspond to individual cells. A mean delta (Δ) value is indicated to represent the drop in YFP lifetime resulting from the pairing of PRRT2-YFP and mCherry versus the pairing of PRRT2-YFP and PRRT2-mCherry. Empty red circles are used to depict mCherry, filled red circles are used to depict PRRT2-mCherry (either WT or G305W), and filled yellow circles are used to depict PRRT2-YFP (either WT or G305W). Error bars indicate standard deviation. Statistical analysis was performed using one-way ANOVA. *, p<0.05; **, p<0.001. Au, arbitrary units. Ps, picoseconds.

It will also be important to assess how oligomerization-defective mutants of IFITM3 (G95L) impact membrane curvature, another reported consequence of ectopic IFITM3 expression in cells (*Li et al., 2013*). It has been shown that protein oligomerization of the transmembrane protein Mic10 is essential for its capacity to induce curvature in mitochondrial membranes (*Barbot et al., 2015*). Furthermore, changes in membrane order are often accompanied by changes in membrane curvature (*Vanni et al., 2014*; *Rangamani et al., 2014*; *Golani et al., 2019*). It is possible that these two alterations to host membranes underlie the restriction of virus fusion by IFITM3. Moreover, amphipathic helices are characterized to interact with and bend membranes (*Giménez-Andrés et al., 2018*). Here, we show for the first time that the amphipathic helix of IFITM3 is critical for the membrane order enhancement by IFITM3. Since we also show that glycine-95 of the GxxxG motif is also required for membrane order enhancement, our data may suggest that oligomerization 'activates' the membrane deforming activity of the amphipathic helix, and as a result, its antiviral potential. It is possible that local insertion of multiple amphipathic helices into stretches of membrane is required for inhibition of virus fusion, and IFITM3 oligomers provide a means to fulfill that requirement.

In addition to mediating dimerization or oligomerization of transmembrane proteins (homomultimerization), GxxxG motifs have also been described to affect the propensity for interaction with other proteins (heteromultimerization) (*Teese and Langosch, 2015*; *Faingold et al., 2012*). Therefore, the GxxxG motif may also govern which proteins IFITM3 interacts with and to what extent. IFITM3 has been described to bind with IFITM1 and IFITM2, and it is interesting to consider how IFITM heteromultimers may contribute to antiviral protection of the cell (*John et al., 2013*).

Furthermore, other host proteins have been described to interact, directly or indirectly, with IFITM3. This list includes cholesterol trafficking regulator VAPA and the metalloproteinase ZMPSTE24, two proteins that have been described as essential cofactors for the antiviral effects of IFITM3 (*Amini-Bavil-Olyaee et al., 2013*; *Fu et al., 2017*). Since the former is associated with the tendency for IFITM3 to cause cholesterol overload in endosomes, the G95L loss-of-function mutant could be used to rule in or rule out VAPA and cholesterol as players in the antiviral mechanism of IFITM3.

While glycine-95 is critical for the oligomerization and anti-fusion activity of IFITM3, we show that the homologous site in PRRT2, glycine-305, regulates its oligomerization as well. The naturally occurring G305W/R mutations found in patients with neurological dysfunction are known to disrupt PRRT2 activity, and our results here provide novel insight into how loss-of-function occurs. Therefore, the presence of a shared GxxxG motif in IFITM3, PRRT2, and some other CD225 family members suggests that an ancestral CD225-containing protein performed an unknown function that required oligomerization. Interestingly, the GxxxG motif is not intact in human IFITM5 and IFITM10. IFITM5 is involved in bone formation and exhibits some antiviral activity when expressed ectopically (*Huang et al., 2011*), while the function of IFITM10 is unknown (*Smith et al., 2014*). We wonder whether all CD225 proteins play roles in regulating membrane fusion processes in cells—only time and further experiments will tell. However, just as the GxxxG motif in IFITM3 may mediate both homo- and heteromultimerization, it has been reported that PRRT2 interacts with cellular fusogens known as soluble N-ethylmaleimide-sensitive factor attachment protein receptors (SNAREs) in a glycine-305-dependent manner (*Coleman et al., 2018*). Our findings raise the possibility that oligomerization of CD225 proteins results in alteration of protein architecture and the display of novel docking sites for protein-protein interactions, allowing for expansion of their functional repertoire.

## Materials and methods

### Key resources table

| Reagent type (species) or resource | Designation | Source or reference | Identifiers | Additional information |
|---|---|---|---|---|
| Cell line (*Homo-sapiens*) | HEK293T | ATCC | CRL-3216 RRID:CVCL_0063 | |
| Cell line (*Homo-sapiens*) | TZM-bl | NIH AIDS Reagent Resource | 8129–442 RRID:CVCL_B478 | |
| Strain, strain background (*Influenza A Virus*) | A/PR/8/34 (H1N1) | Charles River Laboratories | 10100781 | Clarified allantoic fluid |
| Recombinant DNA reagent | pNL4-3 (plasmid) | NIH AIDS Reagent Resource | 114 | |
| Recombinant DNA reagent | pNL4-3ΔEnv (plasmid) | Eric O. Freed | | |
| Recombinant DNA reagent | pMD2.G (VSV-G) (plasmid) | Addgene | 12259 RRID:Addgene_12259 | |
| Recombinant DNA reagent | pCMV4-BlaM-Vpr (plasmid) | Addgene | 21950 RRID:Addgene_21950 | |
| Recombinant DNA reagent | pReceiver-YFP-IFITM3 and mutants (plasmid) | This paper | | YFP fused to IFITM3 at amino terminus |
| Recombinant DNA reagent | pReceiver-mCherry-IFITM3 and mutants (plasmid) | This paper | | mCherry fused to IFITM3 at amino terminus |
| Recombinant DNA reagent | pReceiver-PRRT2-YFP and mutants (plasmid) | This paper | | YFP fused to PRRT2 at carboxy terminus |

*Continued on next page*

*Continued*

| Reagent type (species) or resource | Designation | Source or reference | Identifiers | Additional information |
|---|---|---|---|---|
| Recombinant DNA reagent | pReceiver-PRRT2-mCherry and mutants (plasmid) | This paper | | mCherry fused to PRRT2 at carboxy terminus |
| Recombinant DNA reagent | pQCXIP-FLAG-IFITM3 and mutants (plasmid) | *Compton et al., 2014* and this paper | | FLAG fused to IFITM3 at amino terminus |
| Recombinant DNA reagent | EEA1-GFP (plasmid) | Addgene | 42307 RRID:Addgene_42307 | |
| Commercial assay or kit | LiveBLazer FRET-B/G Loading Kit with CCF2-AM | Thermo Fisher | K1032 | |
| Chemical compound, drug | Amphotericin B | Sigma | C4951 | |
| Chemical compound, drug | FliptR | Spirochrome | CY-SC020 | |
| Chemical compound, drug | Laurdan | Invitrogen | D250 | |
| Chemical compound, drug | Cholesterol, water-soluble | Sigma | A2942 | |
| Chemical compound, drug | Methyl-beta-cyclo-dextrin | Sigma | C4555 | |
| Antibody | Anti-IAV NP mouse monoclonal | Abcam | AA5H | 1:500 (flow) |
| Antibody | Anti-p24 CA mouse monoclonal | NIH AIDS Reagent Resource | 3537 | 1:1000 (WB) |
| Antibody | Anti-p24 CA [KC57-FITC] mouse monoclonal | BD | CO6604665 | 1:500 (flow) |
| Antibody | Anti-CD63 [MX-49.129.5] mouse monoclonal | Santa Cruz Biotechnology | sc-5275 | 1:400 (IF) |
| Antibody | Anti-FLAG [M2] mouse monoclonal | Sigma | F1804 | 1:1000 (WB) 1:500 (flow) |
| Antibody | Anti-IFITM3 [EPR5242] rabbit monoclonal | Abcam | ab109429 | 1:1000 (WB) 1:200 (IF) |
| Antibody | Anti-Env gp120b sheep polyclonal | NIH AIDS Reagent Resource | 288 | 1:1000 (WB) |
| Antibody | Anti-Env gp41 [2F5] human monoclonal | NIH AIDS Reagent Resource | 1475 | 1:1000 (WB) |
| Antibody | Anti-Actin [C4] mouse monoclonal | Santa Cruz Biotechnology | sc-47778 | 1:1000 (WB) |
| Antibody | Anti-c-Myc rabbit monoclonal | Sigma | C3956 | 1:1000 (WB) |

## Sequence retrieval and alignments

Protein sequences for CD225 proteins (including IFITM proteins, PRRT2, and TUSC5) were retrieved from UniProt and multi-sequence alignments were performed with ClustalX.

## Cell lines and plasmids

HEK293T (ATCC: CRL-3216) and TZM-bl (NIH AIDS Reagent Resource: 8129) and any derivatives produced in this study were cultivated at 37°C and 5% $CO_2$ in DMEM (Gibco) complemented with 10% fetal bovine serum (Hyclone) and 1% penicillin-streptomycin (Gibco) and regularly passaged with the aid of Trypsin-EDTA 0.05% (Gibco). All cell lines tested negative for mycoplasma. Retroviral pQCXIP vectors encoding IFITM3 fused with amino-terminal FLAG were previously described (*Compton et al., 2016*; *Jia et al., 2012*). Retroviral pQCXIP vectors encoding IFITM3 fused with amino-terminal myc were produced by appending myc by PCR and cloning into BamH1/EcoR1 sites. pReceiver constructs encoding IFITM3 tagged with YFP or mCherry at the amino terminus were produced by Genecopeia. IFITM3 tagged with YFP or mCherry internally after residue 40 were produced by Integrated DNA Technologies (in these constructs, YFP or mCherry flanked on both sides by a flexible linker (GGGSGG) was inserted after residue 40 and residues 41 and 42 of IFITM3 were deleted, as performed in *Suddala et al., 2019*). PRRT2 tagged with carboxy-terminal YFP or mCherry were produced by Integrated DNA Technologies. Mutations in IFITM3 and the G305W mutation in PRRT2 were introduced by site-directed mutagenesis (QuikChange Lightning) or by ligation-independent cloning. HEK293T cell lines stably expressing pQCXIP plasmids were produced by transfecting 250,000 cells in a 12-well plate with 0.8 μg DNA using Lipofectamine2000 (Invitrogen) and selecting with puromycin at a concentration of 10 μg/mL for at least 2 weeks.

## Virus productions, infections, and virus-cell fusion assay

Influenza A Virus [A/PR/8/34 (PR8), H1N1] supplied as clarified allantoic fluid was purchased from Charles River Laboratories. Infectious virus titers were calculated using a flow cytometry-based method in HEK293T cells (*Grigorov et al., 2011*), and infections were performed as follows: HEK293T cells (either stably expressing or transiently transfected with 1.5 μg empty pQCXIP or IFITM3 WT or the indicated mutant fused with amino terminal FLAG) were seeded in 24-well plates (50,000 per well) overnight and overlaid with indicated amounts of virus diluted in 225 μL of complete DMEM for approximately 18 hr. Cells were washed with 1X PBS, detached with Trypsin-EDTA, fixed/permeabilized with Cytofix/Cytoperm (BD), immunostained with anti-IAV NP (AA5H; Abcam), and analyzed on a LSRFortessa flow cytometer (BD). Replication-incompetent HIV-1 pseudotyped with IAV WSN HA and NA was produced by transfecting HEK293T with 2 μg pR8ΔEnv, 1 μg pcRev (NIH AIDS Reagent Resource: 11415), 3 μg Gag-GFP, 1.5 μg of hemagglutinin, and 1.5 μg of neuraminidase from IAV WSN strain, H1N1 (gifts from G. Melikyan). Replication-incompetent HIV-1 pseudotyped with VSV-G for virus-cell fusion assays was produced by transfecting HEK293T with 15 μg pNL4-3ΔEnv, 5 μg pCMV4-BlaM-Vpr, and 5 μg pMD2.G (VSV-G). Replication-competent HIV-1 was produced by transfecting HEK293T with 15 μg pNL4-3 and 5 μg empty pQCXIP or pQCXIP encoding IFITM3 WT or the indicated mutant fused with amino-terminal FLAG. Transfections were performed using the calcium-phosphate method. Briefly, 6 million HEK293T were seeded in a T75 flask. Plasmid DNA was mixed with sterile $H_2O$, $CaCl_2$, and Tris-EDTA (TE) buffer, and the totality was combined with Hepes-buffered saline (HBS). The transfection volume was added dropwise, and cells were incubated at 37°C for 48 hr. Supernatants were clarified by centrifugation, passed through a 0.45 μm filter, and concentrated by ultracentrifugation through a 20% sucrose cushion at 25,000 x g for 1 hr at 4°C. Lentivirus titers were measured using an HIV-1 p24 ELISA kit (XpressBio). To measure infectivity of virus supernatants, 50 ng p24 equivalent volumes were added to TZM-bl cells and cells were fixed/permeabilized with Cytofix/Cytoperm (BD) at 48 hr post-infection, immunostained with anti-Gag KC57-FITC (BD), and analyzed by flow cytometry. To measure virus-cell fusion, 100–300 ng p24 equivalent HIV-1-VSV-G produced with pCMV4-BlaM-Vpr was added to HEK293T cells stably expressing empty pQCXIP, IFITM3 WT or the indicated mutant for 2.5 hr at 37°C. Cells were washed with $CO_2$-independent medium and incubated with CCF2/AM mix containing probenecid for one hour at room temperature in the dark. Cells were washed with cold 1X PBS, fixed/permeabilized with Cytofix/Cytoperm (BD), and analyzed on a LSRII flow cytometer.

## Confocal immunofluorescence microscopy

HEK293T cells were seeded in μ-slide eight-well chambers (Ibidi) (30,000 per well) overnight and transfected with 0.2 μg EEA1-GFP (Addgene: 42307) and 0.2 μg empty pQCXIP or pQCXIP-WT IFITM3 or the indicated mutant fused with amino-terminal FLAG using Lipofectamine2000 (Invitrogen). At 48 hr post-transfection, cells were fixed/permeabilized with Cytofix/Cytoperm and immunostained with anti-CD63 antibody (MX-49, sc-5275; Santa Cruz Biotechnology) and anti-IFITM3 antibody (EPR5242, ab109429; Abcam). Cells were imaged using the Leica TCS SP8 confocal microscope with a 63X objective and oil immersion and analysis was performed in Fiji (ImageJ). Colocalization analysis was performed with the Colocalization tool in Imaris (Bitplane) as follows: a region of interest was manually created to single out cells which were positive for EEA1-GFP and colocalization between IFITM3/EEA1-GFP and IFITM3/CD63 was measured in 3D using confocal stacks of 20–30 optical sections.

## Immunoprecipitation, SDS-PAGE, and western blot analysis

HEK293T cells were transfected with 1.5 μg empty pQCXIP or a combination of 0.75 μg pQCXIP-FLAG-IFITM3 and 0.75 μg pQCXIP-myc-IFITM3 (encoding either WT or G95L). At 48 hr post-transfection, cells were lysed in a buffer containing 0.5% IPEGAL (Sigma), 50 mM Tris (pH 7.4), 150 mM NaCl, and 1 mM EDTA. Immunoprecipitation was performed using anti-FLAG M2 magnetic beads (Sigma) for a period of 3 hr with rotation at 4℃. Magnetic beads were isolated with a DynaMag-2 magnet (Thermo Fisher) and washed prior to addition of 1X NuPAGE LDS sample buffer and 1X NuPAGE Sample Reducing Reagent (Invitrogen). The non-immunoprecipitated whole cell lysates and immunoprecipitated fractions were heat denatured at 90℃ for 15 min and 12 μL and 5 μL, respectively, were loaded into Criterion XT 12% Bis-Tris polyacrylamide gels (Bio-Rad) for SDS-PAGE using NuPAGE MES SDS Running Buffer (Invitrogen). Proteins were transferred Amersham Protran Premium Nitrocellulose Membrane, pore size 0.20 μm (GE Healthcare). Membranes were blocked with Odyssey blocking buffer in PBS (Li-COR) and incubated with anti-FLAG M2 (F1804; Sigma) and anti-c-Myc (C3956; Sigma). Secondary antibodies conjugated to DyLight 800 or 680 (Li-COR) and the Li-COR Odyssey imaging system were used to reveal specific protein detection. Images were analyzed and assembled using ImageStudioLite (Li-COR). To compare the steady-state expression levels of WT IFITM3 and mutants, HEK293T cells were transfected with 1.5 μg of empty pQCXIP or pQCXIP-FLAG-IFITM3 (encoding WT or the indicated mutants) and expression was measured using anti-FLAG M2 and analysis by flow cytometry or western blot analysis. For the former, cells were fixed/permeabilized with Cytofix/Cytoperm (BD) at 48 hr post-transfection, immunostained with anti-FLAG M2 (F1804; Sigma) and analyzed on a LSRFortessa (BD). For the latter, cells were lysed and separated by SDS-PAGE as indicated above and immunoblotting was performed with anti-FLAG M2 (F1804; Sigma) and anti-actin (C4, sc-47778; Santa Cruz Biotechnology). For blotting of HIV-1 proteins in cell lysates and concentrated virus supernatants, cell/virion lysis was performed with radioimmunoprecipitation (RIPA) buffer (Thermo Fisher) supplemented with Halt Protease Inhibitor mixture EDTA-free (Thermo Fisher) or 0.01% Triton-X (Sigma), respectively, and lysates were complemented with 1X NuPAGE LDS sample buffer and 1X NuPAGE Sample Reducing Reagent (Invitrogen) prior to heat denaturation at 90℃ for 15 min. Samples were migrated and transferred as indicated above and immunoblotting was performed with the following antibodies: anti-p24 CA (NIH AIDS Reagent Resource: 3537), anti-gp120b (NIH AIDS Reagent Resource: 288), anti-gp41 2F5 (NIH AIDS Reagent Resource: 1475), anti-actin (C4, sc-47778; Santa Cruz Biotechnology), and anti-FLAG M2 (F1804; Sigma).

## Blue native PAGE

HEK293T cells were seeded in 6-well plates (750,000 per well) overnight and transfected with 1 μg empty pQCXIP or pQCXIP-FLAG-IFITM3 (either WT or G95L). At 48 hr post-transfection, cells were lysed with 1X NativePAGE Sample Buffer (Invitrogen) containing 1% digitonin. Lysates were mixed with 5% Coomassie G-250 at a volume to volume ratio of 20:1. Approximately 20 μg of protein per sample was loaded into a NativePAGE Novex 4–16% Bis-Tris polyacrylamide gel (Invitrogen) according to manufacturer's instructions. A NativeMark Unstained Protein Standard was loaded as reference ladder. Following PAGE, proteins were transferred to Immobile FL PVDF membrane (EMD Millipore) and immunoblotting was performed with anti-FLAG M2 (F1804; Sigma). A secondary

antibody conjugated to DyLight 800 or 680 (Li-COR) and the Li-COR Odyssey imaging system were used to reveal specific protein detection. Images were analyzed and assembled using ImageStudio-Lite (Li-COR).

## FRET and FLIM for oligomerization studies

HEK293T cells were seeded in μ-slide eight well chambers (Ibidi) (50,000 per well) overnight and transiently co-transfected with 0.25 μg IFITM3-YFP and 0.25 μg mCherry or 0.25 μg IFITM3-YFP and 0.25 μg IFITM3-mCherry using TransIT-293 (Mirus). In parallel experiments, pairs of plasmids encoding PRRT2-YFP and PRRT2-mCherry were co-transfected. Living cells in Fluorobrite DMEM (Gibco) were imaged with a Zeiss LSM 780 confocal microscope using a 63X objective and oil immersion. To assess FRET, donor YFP fluorescence was detected with a gallium arsenide phosphide photomultiplier tube (GaAsP PMT) with a 520–550 nm emission window following excitation by a 514 nm laser. Acceptor mCherry fluorescence was detected with a GaAsP PMT detector with a 570–615 nm emission window following excitation by a 561 nm laser. FRET signal (acceptor mCherry fluorescence triggered by excitation of donor YFP) was collected with a GaAsP PMT detector with 570–615 nm emission window after excitation with a 514 nm laser. At least 50 cells per condition were examined in each experiment. Each cell was assigned a FRET index calculated using the FRET and colocalization analyzer plugin for Fiji (ImageJ). FLIM analysis of donor YFP was performed by excitation with a 950 nm two-photon, pulsed laser (Coherent) tuned at 80 MHz with single photon counting electronics (Becker Hickl) and detection with a HPM-100–40 module GaAsP hybrid PMT (Becker Hickl). Analysis was limited to cells exhibiting 250–1000 photons per pixel to mitigate the effects of photobleaching and low signal to noise ratio. SPCImage NG software (Becker Hickl) was used to acquire the fluorescence decay of each pixel, which was deconvoluted with the instrument response function and fitted to a Marquandt nonlinear least-square algorithm with two exponential models. The mean fluorescence lifetime was calculated as previously described (*Sun et al., 2011*) using SPCImage NG. At least 30 cells per condition were analyzed in each experiment.

## Number and brightness analysis

HEK293T cells were seeded in μ-slide eight-well chambers (Ibidi) (50,000 per well) overnight and transiently with 0.50 μg IFITM3-mCherry using TransIT-293 (Mirus). Living cells in Fluorobrite DMEM (Gibco) were imaged with a Zeiss LSM 780 confocal microscope using a 63X objective and oil immersion. Regions of interest were limited to portions of cells which were immobile and which focused on plasma membrane fluorescence (intracellular signal from vesicular membranes was excluded). The axial position of a specimen during acquisition was stabilized using the Adaptive Focus Control module. mCherry was excited with a 561 nm laser and detected with a 570–615 emission window. For each cell, 100 frames were acquired at a frame rate of 0.385 frames per second with a 9.75 μs pixel dwell time and pixel size of 151.38 nm. Images were always acquired at 256 × 256 pixels such that the pixel size remained three to four times smaller than the volume of the point-spread function. Photobleaching of fluorescent proteins during data acquisition was corrected using a detrending algorithm (*Nolan et al., 2017*). Twenty cellular regions were examined per condition. Pixel-by-pixel brightness values were calculated in Fiji (ImageJ).

## FLIM for study of membrane order with FliptR

HEK293T stably expressing empty pQCXIP or pQCXIP-WT IFITM3 or the indicated mutant were seeded in μ-slide eight-well chambers (Ibidi) (50,000 per well) overnight and stained with 1 μM FliptR (Spirochrome) for 5 min according to the manufacturer's protocol. Imaging was performed with a 63X objective under oil immersion on a Leica SP8-X-SMD confocal microscope. When indicated, cells were treated with 100 μg/mL soluble cholesterol (C4951, Sigma), 5 mM methyl-cyclo-beta-dextrin (MßCD) (C4555, Sigma), or 1 μM Amphotericin B (A2942, Sigma) for 1 hr prior to addition of FliptR and imaging. Fluorescence was detected by hybrid external detectors in photon counting mode following excitation by a 488 nm pulsed laser turned to 20 MHz with single photon counting electronics (PicoHarp 300). Analysis was limited to cells with at least 250–1000 photons per pixel to mitigate the effects of photobleaching and low signal-to-noise ratio. Fluorescence decay of each pixel in FliptR-stained cells was acquired by Symphotime 64 software (Picoquant), and deconvoluted with the instrument response function and fitted to a Marquandt nonlinear least-square algorithm with two

exponential models. The mean fluorescence lifetime (τ), in addition to individual component lifetimes (long and short lifetimes, $\tau_1$ and $\tau_2$), were calculated using Symphotime. At least 30 cells per condition were analyzed in each experiment.

## Laurdan staining for study of membrane order

The membrane probe Laurdan (6-dodecanoyl-2-dimethylamino naphthalene, D250, Invitrogen) was dissolved in DMSO to create a stock solution of 9.19 mM. HEK293T cells were incubated with 1.8 µM Laurdan for 1 hr at 37°C. Generalized polarization (GP) is a ratiometric method which is used to quantitatively report membrane order in living cells. The GP value is calculated as follows:

$$Generalized\,polarization = \frac{I_{blue} - I_{green}}{I_{blue} + I_{green}}$$

where $I_{blue}$ and $I_{green}$ are the fluorescence intensities emitted at 440 nm and 490 nm, respectively. Conventionally, 440 nm and 490 nm are the emission maximums for ordered lipid and disordered lipid bilayers, respectively. Images were acquired with a LSM780 (Zeiss) laser scanning microscope using a 63X oil immersion objective, coupled to a two-photon Ti:Sapphire laser (Coherent) tuned to 780 nm and 80 MHz. An SP 760 nm dichroic filter was used to separate laser light from fluorescence signal. The fluorescence signal was acquired from 416 nm to 474 nm for the blue channel and from 475 nm to 532 nm for the green channel using the GaAsP PMT detectors of the LSM780. GP values were calculated in Fiji (ImageJ) as previously described (*Rentero et al., 2019*). GP values were calculated for each cell of interest and the numeric difference in GP values for the whole cell are presented normalized to empty vector (set to 0).

## Acknowledgements

We thank Guoli Shi, Stephen Lockett, and the Optical Microscopy and Image Analysis Laboratory of the National Cancer Institute, Center for Cancer Research for providing technical support during the acquisition and analysis of microscopy data.

## Additional information

### Funding

| Funder | Grant reference number | Author |
|---|---|---|
| National Institutes of Health | Intramural Research Program | Kazi Rahman<br>Charles A Coomer<br>Saliha Majdoul<br>Selena Y Ding<br>Alex A Compton |
| European Research Council | ERC-2019-CoG-863869 FUSION | Sergi Padilla-Parra |

The funders had no role in study design, data collection and interpretation, or the decision to submit the work for publication.

### Author contributions

Kazi Rahman, Conceptualization, Formal analysis, Supervision, Validation, Investigation, Visualization, Methodology, Writing - review and editing; Charles A Coomer, Data curation, Software, Formal analysis, Validation, Investigation, Visualization, Methodology, Writing - review and editing; Saliha Majdoul, Formal analysis, Investigation, Writing - review and editing; Selena Y Ding, Investigation; Sergi Padilla-Parra, Software, Supervision, Funding acquisition, Methodology, Writing - review and editing; Alex A Compton, Conceptualization, Data curation, Formal analysis, Supervision, Funding acquisition, Validation, Investigation, Visualization, Methodology, Writing - original draft, Writing - review and editing

## Author ORCIDs

Kazi Rahman (iD) https://orcid.org/0000-0003-2986-0007
Charles A Coomer (iD) https://orcid.org/0000-0003-2523-6485
Saliha Majdoul (iD) http://orcid.org/0000-0002-0530-6354
Selena Y Ding (iD) http://orcid.org/0000-0002-4413-644X
Sergi Padilla-Parra (iD) http://orcid.org/0000-0002-8010-9481
Alex A Compton (iD) https://orcid.org/0000-0002-7508-4953

## Decision letter and Author response

Decision letter https://doi.org/10.7554/eLife.58537.sa1
Author response https://doi.org/10.7554/eLife.58537.sa2

## Additional files

### Supplementary files

- Transparent reporting form

### Data availability

All data generated or analysed during this study are included in the manuscript and supporting files.

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
