## [Decision Letter]

**Acceptance summary:**

Interferon induced transmembrane (IFITM) proteins are a subgroup of the dispanin (CD225) domain-containing protein family. IFITMs inhibit the replication of a broad range of enveloped viruses, primarily by inhibiting membrane fusion. The authors identify a motif in the dispanin domain that mediates IFITM3 oligomerization and show that oligomerization correlates with IFITM3-mediated reduction in membrane fluidity and viral restriction. Significantly, a naturally occurring mutation in a similar motif in another dispanin family protein, proline rich transmembrane protein 2 (PRRT2), a neuron-specific regulator of neurotransmitter release, is linked to paroxysmal neurological disorders.

**Decision letter after peer review:**

[Editors’ note: the authors submitted for reconsideration following the decision after peer review. What follows is the decision letter after the first round of review.]

Thank you for submitting your work entitled "Homology-guided identification of a conserved motif linking the antiviral functions of IFITM3 to its oligomeric state" for consideration by *eLife*. Your article has been reviewed by three peer reviewers, including Mark Marsh as the Reviewing Editor and Reviewer #1, and the evaluation has been overseen by a Senior Editor. The following individual involved in review of your submission has agreed to reveal their identity: Camilla Benfield (Reviewer #3).

Our decision has been reached after consultation between the reviewers. Based on these discussions and the individual reviews below, we regret to inform you that your work will not be considered further for publication in *eLife*.

While the reviewers all acknowledge that the mechanisms underlying IFITM inhibition of virus membrane fusion remain unclear, the reviewers point out fundamental concerns with the constructs you have used, i.e. the miss-sorting and loss of functional activity seen by other labs using FP-tagged IFITM proteins. The concerns of the reviewers are detailed below and I hope you find these useful as you develop the work.

Reviewer #1:

Interferon induced transmembrane (IFITM) proteins inhibit the entry of a broad range of enveloped viruses. Focusing mainly on IFITM3, work from a number of labs has suggested that IFITM expression influences membrane stiffness/order, thereby inhibiting viral fusion. Although generally accepted, the mechanism through which IFITMs in general, and IFITM3 in particular, influence membrane stiffness is not fully understood. The corresponding author has previously shown that an amphipathic helix in the conserved CD225/dispannin domain, which is believed to insert into the cytoplasmic leaflet of membranes containing IFITM3, is required for antiviral activity. In this paper, the authors show that a CD225 domain associated GxxxG motif, that in other proteins is referred to as a “glycine zipper”, is required for IFITM3 oligomerization and antiviral activity. Moreover, they show that expression of IFITM3 increases membrane order and that IFITM3 proteins containing mutations of the glycines in the GxxxG motif show reduced oligomer formation and reduced impact on membrane order. Interestingly, they find that another CD225 domain containing protein (PRRT2), that regulates synaptic vesicles, also appears to form oligomers and that disease associated mutations of the PRRT2 GxxxG motif also impact on oligomer formation.

Overall this is an interesting study that supports the notion that IFITM3 inhibits virus fusion by modulating the stiffness of membranes, and that this modulation is associated with the oligomerization mediated at least in part through the GxxxG motif. Overall, I am supportive of publication though I have some concerns with aspects of the data that the authors need to consider.

Figure 2: All data is expressed as relative numbers; the authors should give actual numbers for at least some of the infectivity experiments. Do the authors know why IFITM3 restriction of flu is less effective in stable transfectants that transients? Is there a difference in the level of IFITM3 expression? If so, how does this impact on experiments measuring IFITM3 oligomerization and membrane order? These latter experiments are all done with transiently expressed IFITM3.

Figure 2SB; Previous published work has indicated the bulk of IFITM3 is associated with endosomes. Here it seems that most of the expressed IFITM3-FP is at the cell surface, and very little is in endosomal compartments. Why is this? Weston et al. (PLoS One 2014) previously showed that C terminal tags on IFITM proteins, which in the case of IFITM3 would be exposed to the endosomal lumen, could be proteolytically cleaved. Could the FP tags used here be subject to cleavage in endosomes?

Subsection “Glycine-95 is important for the antiviral functions of IFITM3”; Can the authors explain why the use of an IAV/HIV pseudotype virus reports on early entry events?

Figure 3; In contrast to the distribution of IFITM3 shown in Figure 3, most of the FRET signal reporting oligomerization appears to be on intracellular membranes. Are the FP-tagged proteins trafficking differently, or is oligomerization associated with specific sub-cellular compartments?

Subsection “GxxxG regulates oligomerization of IFITM3 in living cells and bulk lysates”; Can the authors explain what they mean by “diffusion in and out of a static membranous compartment (the plasma membrane)”. The plasma membrane is not a static compartment and how would IFITM3-mCherry be diffusing in and out of this compartment?

Is it valid to compare the intensity of pixels in a 2D array of membrane-associated IFITM3-mCherry with cytoplasmic mCherry?

Reviewer #2:

This is a beautifully written manuscript that describes a functional oligomerization motif within the important innate antiviral immunity protein IFITM3. Despite the excellent writing, the study is generally limited in scope as it essentially boils down to characterization of a single point mutant of IFITM3. Novelty of the paper is also low as the residue in question (G95) has already been shown to be important for IFITM3 activity (John et al., 2013). Furthermore, most of the conclusions of the manuscript are also already known, including that IFITM3 dimerization is essential for activity (John et al., 2013), that IFITM3 increases membrane stiffness (Li, PLOS Path, 2013), and that antiviral function of IFITM3 is countered by Amphotericin (Lin, Cell Reports, 2013). There are also concerns with some of the reagents and much of the data presentation.

1) Fluorescent protein fusions of IFITM3 have been generated and tested by multiple laboratories and none were found to be active regardless of the termini at which the protein was fused or the fluorescent protein that was used. This was such an impediment to live cell imaging of IFITM3 that the top labs in the field have gone to extraordinary measures to visualize IFITM3 in live cells (Spence et al., 2019; Peng, J Am Chem Soc, 2016; Suddala et al., 2019). It is not possible that the fluorescent fusion proteins used in this paper are antivirally active as presented in Figure 3.

2) All of the virus infectivity data is presented as normalized infectivity, and no raw data flow cytometry plots are shown. For influenza infections, an MOI of 0.1 is used. Thus, the maximum % infection would be at most ~7%, providing a minimal dynamic range for observing antiviral effects of IFITM3 or for comparing IFITM3 mutants. This minimal dynamic range, along with the strange results with fluorescent fusions, coupled with unnecessary obfuscation of the primary data, overall decreases confidence in some of the results.

Reviewer #3:

Previously, F75 and F78 were thought to link oligomerisation to antiviral restriction, but recently this was refuted by Winkler et al., 2019, who found no effect of these residues on IFITM3-IFITM3 interactions. Since most previous work used these F75/78 mutants, discovering the determinants of IFITM3 oligomerisation and its relation to antiviral activity remains important and not yet answered.

The authors report that Gly-95 of human IFITM3 reduces restriction, oligomerisation and membrane rigidification by IFITM3.

However, aspects of these conclusions need further support, as I have detailed below. I also feel that the novelty and conceptual advance is likely not sufficient for publication by *eLife*, but rather incremental. IFITM3 has already been shown to increase membrane order (Li et al., 2013) and G95R was previously shown to attenuate restriction of IAV (John et al., 2013).

1) Do the authors, as per the title, link the motif or AA reside 95 to oligomerisation/restriction?, vast majority of data pertains to G95L – see below.

2) The Introduction states that the GXXXG motif is also known as a glycine zipper whereas my understanding is that the most significant glycine zipper sequence patterns are (G,A,S)XXXGXXXG and GXXXGXXX(G,S,T), which contain a GXXXG motif (https://www.pnas.org/content/102/40/14278, not referenced in the current paper)

3) I do not feel this is an evolution-guided analysis (title of the first Results section), which to me would imply analysis of evolutionary selection pressures. I am more comfortable with “homology-guided”, as per the title. I also do not follow the logic of the phylogenetic tree of CD225 domains in 1A “indicative of a common ancestry”, all CD225 superfamily proteins have this domain, by definition. Not sure what the tree adds. Authors also say that the GXXXG motif is conserved in vertebrate IFITM3s but (i) So are many other residues/ AA motifs conserved in the highly conserved CD225 domain of vertebrate IFITM3, as the alignment Figure 1D shows, and (ii) surely to make a point about conservation among vertebrates, an alignment where 5 of the 7 mammalian IFITM3 sequences shown are from primates, i.e. the same mammal order, is less informative than an alignment with wider range of mammal or vertebrate species whose IFITM3 has been sequenced. Do IFITM5 and IFITM10 oligomerise, as they lack an intact GXXXG? (not commented on)

4) Although authors say they focused on G95L and G95W mutants for further functional characterisation (subsection “Glycine-95 is important for the antiviral functions of IFITM3”), no localisation data is shown for G95W and for some reason, G95W is omitted from panels D and F in Figure 2, which shows effects on antiviral restriction. Later, oligomerisation is not assessed for G95W (Figures 3 and 4) and yet despite the lower expression levels noted for G91L by transient transfection (S2A), this G91L is reinserted into the analysis of oligomerisation (Figure supp 3B and interpretation lines subsection “GxxxG regulates oligomerization of IFITM3 in living cells and bulk lysates”). The authors do not explain the logic for this “pick and choose” approach.

5) I do not find that the localisation data which is presented for WT and G95L to be convincing (i.e. Figure S2B; IF performed for transient IFITM3 expression in EEA1-GFP overexpressing cells). The authors state their IF shows IFITM3 distributed in early and late endosomes and the PM (subsection “Glycine-95 is important for the antiviral functions of IFITM3”), whereas the staining to me seems very dominantly PM with minimal endosomal staining? Why is there no IF done on the stable cell lines which much of the restriction data in Figure 2 pertain to (panels B, D, E). Further, since numerous studies have shown localisation to be a key feature/correlate of IFITM3 restriction, stronger data confirming “similar subcellular localisation” of the Glycine mutants should be presented (both G95L and G95W).

6) It is known that expression levels of IFITM3 affect its antiviral restriction. For transient transfection experiments, authors provide a single western blot panel (Figure supplementary 2A) which indicates the two Gly-95 mutants express similarly to WT when transiently expressed. However, it would be better to perform densitometric quantification of replicate independent blots, and/ or provide western panels to shown expression in the same experiments as restriction is assessed following transient transfection. For the stable cell lines, FACS analysis is shown (Figure 2A) which indicates that the mutants are less well expressed in this system. First, I think that replicates should have been analysed in order to show whether there is any statistically significant difference between the cell lines. (Currently only the MFI for a single replicate for each cell line is given: WT=814, G95W=590 , G95L=605). Secondly, the authors need to comment on whether this expression level difference might influence the interpretation of the restriction data obtained for the cell lines, and indeed the other phenotypes seen.

7) Authors state that late stage inhibition of HIV-1 entry is regulated by glycine 95 (subsection “Glycine-95 is important for the antiviral functions of IFITM3”), these data are shown in a single bar chart in Figure 2F and a composite supplementary Figure 2C. In supplementary Figure 2C, the authors should explain how the quantification relates to the Western blot panels- presumably all the quantification graphs refer to the right-hand panels of the westerns labelled “virus supernatants”, but this needs to be clarified. Furthermore, could the (arguably small) differences seen for wt vs. glycine mutant in right-hand panel (supernatants) not be a consequence of the differences seen in the left-hand panel (i.e. expression levels in whole cell lysates), the latter potentially due to the effects shown for IFITM3 on HIV protein synthesis (Lee et al., 2018, “IFITM proteins inhibit HIV-1 protein synthesis”)? It is not clear to me that these data wholly support the authors interpretation that late stage inhibition of HIV-1 entry is regulated by glycine 95 and the conclusion that “oligomerization-defective G95L mutant lessened the impact of virion-associated IFITM3 on HIV-1, suggesting that IFITM3 oligomers are needed to maximally reduce virion infectivity”.

8) Figure 3 (on oligomerisation) is confusing, the circular symbols are not explained and I was unclear what the 3 columns represented and whether the labels at the top (mCherry, IFITm3-YFP and IFITM3-mCherry) pertained to the columns (in which case they need to be centered over the columns) or were a legend. Accordingly, I could not interpret these data or the bar charts below the IF which, again, used these undefined circular symbols. The figure legend refers to filled red and filled yellow circles, which I could not see on the figure. I have the same problem interpreting Figure 6 (analagous experiments with PRRT2).

9) The co-IP and native PAGE data lack loading controls (Figure 4) and I was also unclear why i. the heterologous pairs had not been tried for co-IPs experiments ie WT with G95L, ii. there was no replicates or quantitation for the native PAGE (i.e. an equivalent panel as for 4B).

10) Using the FliptR system, authors report that “membrane order enhancement tracks with a functionally competent form of IFITM3 but not with a loss-of-function mutant”. It would have been useful to assess one of the many other loss of function IFITM mutants that have been described in the literature, e.g. the S-palmitoylation defective mutants or an amphipathic helix-lacking mutant (especially since some of the same authors discovered IFITM3's amphipathic helix). Further, while positive and negative controls for membrane order were used (Figure 6—figure supplement 1), it would help strengthen the data on membrane rigidity if authors had used an additional method, especially one that has been used previously to study IFITM3 e.g. Laurdan.

---

## [Author Response]

[Editors’ note: The authors appealed the original decision. What follows is the authors’ response to the first round of review.]

While the reviewers all acknowledge that the mechanisms underlying IFITM inhibition of virus membrane fusion remain unclear, the reviewers point out fundamental concerns with the constructs you have used, i.e. the miss-sorting and loss of functional activity seen by other labs using FP-tagged IFITM proteins. The concerns of the reviewers are detailed below and I hope you find these useful as you develop the work.Reviewer #1:Overall this is an interesting study that supports the notion that IFITM3 inhibits virus fusion by modulating the stiffness of membranes, and that this modulation is associated with the oligomerization mediated at least in part through the GxxxG motif. Overall, I am supportive of publication though I have some concerns with aspects of the data that the authors need to consider.Figure 2; All data is expressed as relative numbers; the authors should give actual numbers for at least some of the infectivity experiments. Do the authors know why IFITM3 restriction of flu is less effective in stable transfectants that transients? Is there a difference in the level of IFITM3 expression? If so, how does this impact on experiments measuring IFITM3 oligomerization and membrane order? These latter experiments are all done with transiently expressed IFITM3.

In order to render the results more transparent, we have incorporated many examples of raw data throughout the manuscript. Regarding IAV infection, we have included raw infection data as measured by FACS in Figure 2C. For clarity, we removed the panel from Figure 2 showing IAV infection in cells transiently expressed IFITM3 constructs. Now, only the results in stably expressing cell lines are shown there, and these results are more than adequate to show that mutation at G91 and G95 impair anti-IAV activity. To compare the expression levels in transiently-transfected and stably-transfected HEK293T cells, please see Figure 2A and 2B and Figure 1—figure supplement 1A and 1B. In the stable cell lines, IFITM3 constructs are expressed uniformly and at higher levels relative to transient transfections. To address the question of how transient vs stable expression of IFITM3 impact oligomerization and membrane order, we have introduced new data measuring oligomerization and membrane order following transient AND stable expression of IFITM3. The antiviral activity of IFITM3 against IAV is found in Figure 2D (stable) and Figure 4—figure supplement 2A and B (transient). The effect of IFITM3 on membrane order is found in Figure 6B (stable) and Figure 6—figure supplement 1E (transient). Importantly, whether IFITM3 was expressed transiently or stably did not impact the antiviral activity/membrane rigidification by IFITM3 WT nor did it impact the loss of those effects by mutation of the GxxxG motif.

Figure 2SB; Previous published work has indicated the bulk of IFITM3 is associated with endosomes. Here it seems that most of the expressed IFITM3-FP is at the cell surface, and very little is in endosomal compartments. Why is this? Weston et al. (PLoS One 2014) previously showed that C terminal tags on IFITM proteins, which in the case of IFITM3 would be exposed to the endosomal lumen, could be proteolytically cleaved. Could the FP tags used here be subject to cleavage in endosomes?

Thank you for raising this important issue regarding the subcellular localization of IFITM3 and its mutants. The data presented in former Supplemental Figure 2B was actually that of IFITM3 tagged with FLAG at the N-terminus, not the C-terminus. Furthermore, the fluorescent protein constructs (mCherry and YFP) were also placed at the N-terminus of IFITM3. As the reviewer points out, Weston et al. previously showed that C-terminal tags could be cleaved and we specifically avoided C-terminal tagging for that reason. Anyway, we have now exhaustively tested whether our FLAG-tagged IFITM3 constructs differ with regards to subcellular localization. We have included new immunofluorescence data in Figure 2—figure supplement 1A (stable expression) and 2B (transient expression) and performed quantitative colocalization analysis between IFITM3 and early endosomal marker EEA1 or late endosomal/MVB marker CD63. This analysis shows that a significant amount (50-60%) of IFITM3 stably expressed in HEK293T is present in early and late endosomes, and this proportion was not significantly altered by mutations in the GxxxG motif. In stable cell lines, IFITM3 WT and mutants were more abundant in CD63+ MVB than they were under transient transfection conditions (Figure 2—figure supplement 1A (stable expression) and 2B (transient expression)). Therefore, these data demonstrate that a general loss of endosomal localization is not responsible for the differential antiviral functions exhibited by IFITM3 WT and GxxxG mutants.

Subsection “Glycine-95 is important for the antiviral functions of IFITM3”; Can the authors explain why the use of an IAV/HIV pseudotype virus reports on early entry events?

We have removed these data from Figure 2 as it was unnecessary. Instead, we highlight our data using HIV-VSV-G incorporating BlaM-Vpr in Figure 2E as a readout for viral access to the cytoplasm (entry/fusion).

Figure 3; In contrast to the distribution of IFITM3 shown in Figure 3, most of the FRET signal reporting oligomerization appears to be on intracellular membranes. Are the FP-tagged proteins trafficking differently, or is oligomerization associated with specific sub-cellular compartments?

The reviewer raises an interesting observation. The data reporting FRET and YFP lifetimes of FP-tagged IFITM3 are now shown in Figure 4A. The distribution of YFP lifetimes (lower row) indicate that YFP signal is present at the cell surface and intracellular compartments (endosomes/other vesicles), suggesting that FP-tagged IFITM3 are not trafficking differently compared to the FLAG-tagged constructs analyzed in Figure 2—figure supplement 1. However, the lowest YFP lifetimes in cells, indicative of sites where FP oligomerization is occurring, are those present in endosomes/vesicles. This is most apparent in the WT condition, where the largest drop in YFP lifetime per cell occurs. Accordingly, the FRET signal is also concentrated in intracellular compartments and less FRET is detectable on the cell surface. These data may suggest that IFITM3 oligomers are mostly localized to intracellular membranes such as endosomes while fewer IFITM3 oligomers are present on the cell surface. Since this point may be relevant to the various functions performed by IFITM3 in health and disease, we now discuss these results; “Importantly, we noticed that the majority of the FRET signal occurring between IFITM3-YFP and IFITM3-mCherry was detected in intracellular endosomes and other membrane vesicles (Figure 4). In contrast, FRET at the cell surface was relatively low. Therefore, IFITM3 oligomers may be more abundant in endosomal membranes and this agrees with our finding that viruses entering cells via fusion with endosomes (such as IAV) are strongly restricted by IFITM3 oligomers.”

Subsection “GxxxG regulates oligomerization of IFITM3 in living cells and bulk lysates”; Can the authors explain what they mean by “diffusion in and out of a static membranous compartment (the plasma membrane)”. The plasma membrane is not a static compartment and how would IFITM3-mCherry be diffusing in and out of this compartment?

We thank the reviewer for pointing out this oversight. We changed the text to read “We restricted our analysis to brightness of IFITM3-mCherry as it trafficked in and out of the plasma membrane over time.”

Is it valid to compare the intensity of pixels in a 2D array of membrane-associated IFITM3-mCherry with cytoplasmic mCherry?

The reviewer brings up a good point. Our “empty mCherry” is cytoplasmic and does not contain transmembrane domains, meaning that it does not represent the ideal negative control for studying the specific interaction between proteins in membranes using Number and Brightness. We added text to this section to underline this caveat to the analysis: “One caveat of this Number and Brightness analysis is that, in contrast to transmembrane proteins, mCherry monomers do not target membranes and are expressed mostly in the cytoplasm. Nonetheless, the fact that IFITM3 WT exhibits a brightness that is roughly two-fold greater than IFITM3 G95L supports the notion that the former exists primarily as a dimer and the latter as a monomer.”

Reviewer #2:This is a beautifully written manuscript that describes a functional oligomerization motif within the important innate antiviral immunity protein IFITM3. Despite the excellent writing, the study is generally limited in scope as it essentially boils down to characterization of a single point mutant of IFITM3. Novelty of the paper is also low as the residue in question (G95) has already been shown to be important for IFITM3 activity (John et al., 2013). Furthermore, most of the conclusions of the manuscript are also already known, including that IFITM3 dimerization is essential for activity (John et al., 2013), that IFITM3 increases membrane stiffness (Li, PLOS Path, 2013), and that antiviral function of IFITM3 is countered by Amphotericin (Lin, Cell Reports, 2013). There are also concerns with some of the reagents and much of the data presentation.

We thank the reviewer for their praise. Concerning the statement that this article “boils down to a single point mutant,” we have added significant amounts of new data to highlight the effects of both G91 and G95 of the GxxxG motif. As a result, data on G91 is now found in Figures 1-4 and the data are less focused on a single mutation. Our new findings indicate that mutation of G95 has a larger impact on protein oligomerization than G91, which justifies our focus on G95 in the last two figures of the paper. We tried to study a double mutant whereby the G91L and G95L mutations were introduced simultaneously into IFITM3, but the polypeptide product was not expressed at detectable levels following transfection into HEK293T (not shown). Overall, our paper describes the functional impact of the GxxxG motif rather than a single point mutation, and this underlines the novelty of our study because the GxxxG motif in IFITM3 has not previously been reported or functionally described.

Additionally, the reviewer has failed to appreciate the other points of novelty in our manuscript:

1) While mutation of G95 was previously shown to impact anti-IAV activity, the reason for loss of function was unknown. Here, we provide the mechanism for loss-of-function.

2) While a previous publication suggested that IFITM3 dimerizes (John et al., 2013), that study implicated F75 and F78 as drivers for oligomerization. Here, we added fresh data confirming that F75/F78 do not play a critical role in oligomerization as measured by our FRET approach (Figure 4—figure supplement 1), which agree with another study utilizing FRET (Winkler et al., 2019). Even though John et al. showed that F75 and F78 are important for antiviral function, the fact that F75 and F78 are dispensable for oligomerization means that the authors did not show that “dimerization is essential for activity.” On the contrary, our results leveraging mutation of the GxxxG motif are the first to indicate that oligomerization is essential for antiviral activity.

3) While we openly agree that previous literature demonstrated that IFITM3 increases membrane stiffness and that Amphotericin B overcomes the antiviral activity of IFITM3, we are the first to directly show that membrane stiffness is directly linked to antiviral function. By using a loss-of-function mutation (G95L), we are the first to show that membrane stiffening is required for antiviral activity. Furthermore, by incorporating Amphotericin B into the same assay, we are the first to show that Amphotericin B negates the impact of IFITM3 on membrane stiffness. Together, these data provide significant novel information about how IFITM3 performs its antiviral activities AND how a drug overcomes these antiviral activities.

4) The study is not “limited in scope,” since our identification of a GxxxG in IFITM3 and other CD225 proteins is indicative of a shared requirement for oligomerization in the diverse membrane fusion processes controlled by the CD225 protein family. We show that mutation of G305W in PRRT2 disrupts its ability to oligomerize, adding fresh insight this mutation that is directly linked to neurological disease in humans.

1) Fluorescent protein fusions of IFITM3 have been generated and tested by multiple laboratories and none were found to be active regardless of the termini at which the protein was fused or the fluorescent protein that was used. This was such an impediment to live cell imaging of IFITM3 that the top labs in the field have gone to extraordinary measures to visualize IFITM3 in live cells (Spence et al., 2019; Peng, J Am Chem Soc, 2016; Suddala et al., 2019). It is not possible that the fluorescent fusion proteins used in this paper are antivirally active as presented in Figure 3.

We understand that fluorescent tags can impede the behavior of membrane proteins in live cell imaging. We are not ignorant of the results achieved by Spence et al. and Suddala et al., which were published during the preparation of our manuscript. We have included new data outlining the oligomerization potential and antiviral activity of IFITM3 constructs madae *a la* Suddala et al.—that is, we introduced the YFP/mCherry tags internally into IFITM3, after codon 40. We then compared the anti-IAV activity of these internally-tagged IFITM3 constructs with our IFITM3 constructs containing N-terminal tags. Our results showed that the IFITM3 constructs containing YFP/mCherry at the N-terminus exhibit more potent antiviral activity than those encoding YFP/mCherry internally. Specifically, our YFP/mCherry placed at the N-terminus resulted in a 20% loss of antiviral function compared to FLAGIFITM3, while YFP/mCherry placed internally at residue 40 resulted in a 40% loss of antiviral function.

An example of raw data and the combined results from three infection experiments is now shown in Figure 4—figure supplement 2A and B. We also went ahead and assessed whether the internally-tagged constructs were competent for FRET as measured by FLIM microscopy. We found the WT versions of these constructs produced FRET and that the G95L mutation abrogated FRET (Figure 4—figure supplement 2C), mirroring our original results obtained using IFITM3 tagged with YFP/mCherry at the N-terminus (Figure 4B and 4C). Therefore, mutation of G95 results in loss of oligomerization in all fluorescently tagged constructs. Our focus on the use of IFITM3 fluorescently tagged with YFP and mCherry at the Nterminus is justified by their more potent antiviral activity. While these results may seem at odds with Suddala et al., they tagged IFITM3 internally with different fluorophores (mNeonGreen and mTFP1) than we did (YFP and mCherry) and the antiviral potential of these different constructs may be impacted by fluorophore choice.

An equally important point to stress here is that we utilized FLAG-IFITM3 and myc-IFITM3 to address oligomerization as well (in Figure 5), in order to confirm our results using fluorescently-tagged IFITM3. Since the reviewer does not question the antiviral function performed by FLAG-IFITM3, our findings with FLAG-IFITM3 and myc-IFITM3 further support that oligomerization is critical for antiviral function.

2) All of the virus infectivity data are presented as normalized infectivity, and no raw data flow cytometry plots are shown. For influenza infections, an MOI of 0.1 is used. Thus, the maximum % infection would be at most ~7%, providing a minimal dynamic range for observing antiviral effects of IFITM3 or for comparing IFITM3 mutants. This minimal dynamic range, along with the strange results with fluorescent fusions, coupled with unnecessary obfuscation of the primary data, overall decreases confidence in some of the results.

We agree that our manuscript would be better appreciated if data obfuscation was minimized. In order to render the results more transparent, we have incorporated many examples of raw data throughout the manuscript. Raw data of flow cytometry results can now be found in Figure 2C (IAV infection of HEK293T stably expressing IFITM3 constructs), Figure 3A (HIV infection of TZMbl cells as a readout of virion infectivity), Figure 1—figure supplement 1E (fusion of HIV-BlaM-Vpr-VSV-G in HEK293T stably expressing IFITM3 constructs), and Figure 4—figure supplement 2A (IAV infection of HEK293T transiently expressing FLAG- and fluorescent protein-tagged IFITM3).

Figure 2C shows raw data for IAV infections performed at an MOI of 0.1. We observe about ~12% infection occurring in HEK293T cells at this MOI, which is similar to the ~7% predicted by the reviewer. However, we have also performed other experiments using higher and lower MOIs, and our conclusions are not impacted. For example, please consider the raw FACS data in Author response image 1:

These data show that varying the MOI from 0.05 to 0.5 does not influence our measurements of antiviral function by IFITM3 WT or mutants. Therefore, we believe our choice to report infection data using an MOI of 0.1 is appropriate.

Reviewer #3:However, aspects of these conclusions need further support, as I have detailed below. I also feel that the novelty and conceptual advance is likely not sufficient for publication by eLife, but rather incremental. IFITM3 has already been shown to increase membrane order (Li et al., 2013) and G95R was previously shown to attenuate restriction of IAV (John et al., 2013).

We thank the reviewer for appreciating the importance of our results with regards to IFITM3 oligomerization. However, the reviewer has failed to appreciate the other points of novelty in our manuscript:

1) While mutation of G95 was previously shown to impact anti-IAV activity, the reason for loss of function was unknown. Here, we provide the mechanism for loss-of-function.

2) While we openly agree that previous literature demonstrated that IFITM3 increases membrane order, we are the first to directly show that membrane order is directly linked to antiviral function. By using a loss-of-function mutation (G95L), we are the first to show that membrane stiffening is required for antiviral activity. Furthermore, by incorporating Amphotericin B into the same assay, we are the first to show that Amphotericin B negates the impact of IFITM3 on membrane stiffness. Together, these data provide significant novel information about how IFITM3 performs its antiviral activities AND how a drug overcomes these antiviral activities.

3) We believe this manuscript is appropriate for publication in *eLife* because of the cross-cutting nature of the data—it is not merely an incremental advance on the function of IFITM3. The basis for our study was actually gleaned from the fact that a non-synonymous SNP (G305W) in PRRT2 had been associated with neurological disease in humans. We decided to study the homologous residue in IFITM3 (G95) and identified a GxxxG motif in IFITM3 and as well as other CD225 proteins, indicating a shared requirement for oligomerization in the diverse membrane fusion processes controlled by the CD225

protein family. Importantly, we show that mutation of G305W in PRRT2 disrupts its ability to oligomerize, adding fresh insight to our understanding of how this mutation results in disease.

1) Do the authors, as per the title, link the motif or AA reside 95 to oligomerisation/restriction?, vast majority of data pertains to G95L – see below.

We have added significant amounts of new data to highlight the effects of both G91 and G95 of the GxxxG motif. As a result, data on G91 is now found in Figures 1-4 and the data are less focused on a single mutation. Our new findings indicate that mutation of G95 has a larger impact on protein oligomerization than G91, which justifies our focus on G95 in the latter figures of the paper. We tried to study a double mutant whereby the G91L and G95L mutations were introduced simultaneously into IFITM3, but the polypeptide product was not expressed at detectable levels following transfection into HEK293T (not shown). Overall, our paper describes the functional impact of the GxxxG motif rather than a single point mutation, and this underlines the novelty of our study because the GxxxG motif in IFITM3 has not previously been reported or functionally described.

2) The Introduction states that the GXXXG motif is also known as a glycine zipper whereas my understanding is that the most significant glycine zipper sequence patterns are (G,A,S)XXXGXXXG and GXXXGXXX(G,S,T), which contain a GXXXG motif (https://www.pnas.org/content/102/40/14278, not referenced in the current paper)

We thank the reviewer for raising this point. Since the GxxxG sequence in IFITM3 does not clearly belong to an extended motif resembling a glycine zipper (GxxxGxxxG or similar), we removed “glycine zipper” from the text. Instead we chose to refer to the motif as a “GxxxG motif, also known as a (small)xxx(small) motif” as described by one of our references (Teese et al., 2015).

3) I do not feel this is an evolution-guided analysis (title of the first Results section), which to me would imply analysis of evolutionary selection pressures. I am more comfortable with “homology-guided”, as per the title. I also do not follow the logic of the phylogenetic tree of CD225 domains in 1A “indicative of a common ancestry”, all CD225 superfamily proteins have this domain, by definition. Not sure what the tree adds. Authors also say that the GXXXG motif is conserved in vertebrate IFITM3s but (i) So are many other residues/ AA motifs conserved in the highly conserved CD225 domain of vertebrate IFITM3, as the alignment Figure 1D shows, and (ii) surely to make a point about conservation among vertebrates, an alignment where 5 of the 7 mammalian IFITM3 sequences shown are from primates, i.e. the same mammal order, is less informative than an alignment with wider range of mammal or vertebrate species whose IFITM3 has been sequenced. Do IFITM5 and IFITM10 oligomerise, as they lack an intact GXXXG? (not commented on)

We agree that the term “evolution-guided” is inappropriate. This statement has been corrected to the preferred language “homology-directed.” We also removed the phylogenetic tree since the common ancestry of the CD225 proteins, specifically that of IFITM3 and PRRT2, is clearly described elsewhere. To strengthen the notion that GxxxG is conserved among vertebrates, we updated Figure 1D to include the additional species: cow, cat, microbat, and sperm whale. With this expanded list of species, the conservation of the GxxxG motif is even more apparent.

Regarding IFITM5 and IFITM10, we introduced additional text in the Discussion stating “Interestingly, the GxxxG motif is not intact in human IFITM5 and IFITM10. It is tempting to speculate that decay of the GxxxG motif is one reason why an antiviral function for these IFITM members is yet to be reported. It is also possible that oligomerization is unimportant for their respective non-antiviral roles in cells: IFITM5 is involved in bone formation, while the function of IFITM10 is unknown.”

4) Although authors say they focused on G95L and G95W mutants for further functional characterisation (subsection “Glycine-95 is important for the antiviral functions of IFITM3”), no localisation data are shown for G95W and for some reason, G95W is omitted from panels D and F in Figure 2, which shows effects on antiviral restriction. Later, oligomerisation is not assessed for G95W (Figures 3 and 4) and yet despite the lower expression levels noted for G91L by transient transfection (S2A), this G91L is reinserted into the analysis of oligomerisation (Figure supp 3B and interpretation lines subsection “GxxxG regulates oligomerization of IFITM3 in living cells and bulk lysates”). The authors do not explain the logic for this “pick and choose” approach.

We agree that the absence of G95W from certain figures made the data appear imbalanced. We have now included additional data on IFITM3 G95W, including results on its expression level, subcellular localization, and antiviral activity. These data can now be found in Figure 2, Figure 1—figure supplement 1, and Figure 2—figure supplement 1. We quantified the protein expression of IFITM3 WT and mutants, including G95W, by flow cytometry and western blotting and performed quantitative and statistical analysis for both methods. Here are some highlights of what we found:

1) G95W lost activity against IAV and VSV-G-driven virus-cell fusion to approximately the same degree as G95L;

2) G95W and G95L exhibit a similar subcellular localization relative to IFITM3 WT, in that all colocalize with the plasma membrane, early endosomes, and late endosomes to similar extents following transient and stable transfection in HEK293T. Our quantitative analysis revealed that about ~50% of IFITM3 resides in endolysosomes, irrespective of the mutations in the GxxxG motif.

Due to the fact that G95W behaves similarly to G95L in terms of its antiviral activity, we did not think it was necessary to explore the effect of both G95L and G95W on oligomerization and membrane order. Overall, we are hopeful that the reviewer will be satisfied with the fact that results for G91L, G95L, and G95W are now presented side-by-side in Figure 2, Figure 1—figure supplement 1, and Figure 2—figure supplement 1.

5) I do not find that the localisation data which is presented for WT and G95L to be convincing ( i.e. Figure S2B; IF performed for transient IFITM3 expression in EEA1-GFP overexpressing cells). The authors state their IF shows IFITM3 distributed in early and late endosomes and the PM (subsection “Glycine-95 is important for the antiviral functions of IFITM3”), whereas the staining to me seems very dominantly PM with minimal endosomal staining? Why is there no IF done on the stable cell lines which much of the restriction data in Figure 2 pertain to (panels B, D, E). Further, since numerous studies have shown localisation to be a key feature/correlate of IFITM3 restriction, stronger data confirming “similar subcellular localisation” of the Glycine mutants should be presented (both G95L and G95W).

In response to this criticism, we completely overhauled our immunofluorescence data. We performed additional experiments whereby IFITM3 and all mutants (G91L, G95L, and G95W) in both transiently and stably transfected HEK293T cells were localized relative to EEA1-GFP and CD63. We performed transfections and stainings in three independent experiments and performed colocalization analysis on Zstacks using Imaris software. These new results can now be found in Figure 2—figure supplement 1. We found that approximately 50% of IFITM3 is present in early/late endosomes following ectopic expression, which is broadly consistent with our previous work studying FLAG-IFITM3 (Compton et al., 2016). We also found that cells stably expressing IFITM3 contain higher levels of IFITM3 in CD63+ late endosomes/MVB, which likely explains why stably expressing cells exhibit more potent antiIAV activity than cells which were transiently transfected. Importantly, there were no statistically significant differences between the degree to which WT, G91L, G95L, and G95W colocalized with early/late endosomes (Figure 2—figure supplement 1). These results suggest that mutations in the GxxxG motif result in loss of function for reasons other than altered subcellular localization.

6) It is known that expression levels of IFITM3 affect its antiviral restriction. For transient transfection experiments, authors provide a single western blot panel (Figure supplementary2 A) which indicates the two Gly-95 mutants express similarly to WT when transiently expressed. However, it would be better to perform densitometric quantification of replicate independent blots, and/ or provide western panels to shown expression in the same experiments as restriction is assessed following transient transfection. For the stable cell lines, FACS analysis is shown (Figure 2 A) which indicates that the mutants are less well expressed in this system. First, I think that replicates should have been analysed in order to show whether there is any statistically significant difference between the cell lines. (Currently only the MFI for a single replicate for each cell line is given: WT=814, G95W=590 , G95L=605). Secondly, the authors need to comment on whether this expression level difference might influence the interpretation of the restriction data obtained for the cell lines, and indeed the other phenotypes seen.

We agree that we did not objectively characterize the relative expression levels of IFITM3 WT and mutants. Therefore, we quantitatively assessed levels of IFITM3 WT, G91L, G95L, and G95W by both flow cytometry and western blotting and under conditions of both transient and stable transfection. In both cases, quantitation and statistical analysis from multiple experiments were performed. For the flow cytometry data, we show an example of flow histograms as well as a combined bar graph representing the mean fluorescence intensities from 3-4 experiments (Figure 2A and 2B for stable transfection, Figure 1—figure supplement 1A and 1B for transient transfection). For the western blotting data, we show an example of a membrane scan of cell lysates produced from a single transfection as well as a combined bar graph representing the mean band intensity from 3 experiments as measured by Odyssey Li-Cor technology (which quantitatively measures secondary antibodies coupled to fluorophores emitting in the infrared range, and fluorescence is directed measured; this is superior to traditional densiometry making use of HRP-coupled secondary antibodies, which is a semi-quantitative technique that requires enhanced chemiluminescence) (Figure 1—figure supplement 1C and D for transient transfection). These data allow the comparison of steady-state protein levels between IFITM3 WT and the mutants in a more convincing and transparent manner. This analysis revealed that G95W was expressed at significantly lower levels than WT in stable cell lines (Figure 2B) and G91L was expressed at significantly lower levels than WT in transient transfected cells (Figure 2—figure supplement 1D). Since, in stable cell lines, G95W is just as antiviral as G95L (Figure 2D and 2E), this difference does not seem to impact function. However, the decreased expression of G91L under transient conditions probably does impact our interpretation of Figure 3B, which has been updated in this revision. Here, the G91L mutation results in a complete loss of activity against HIV-1 in producer cells (manifesting in loss of virion infectivity), and we believe this is due to the combined effects of 1) partial loss of oligomerization for IFITM3 containing G91L, and 2) lower steady state levels of IFITM3 containing G91L. Importantly, we did not detect any significantly different levels of expression between WT and G95L irrespective of the method used (flow cytometry, western blot) or the transfection (transient, stable). Therefore, any functional difference observed between WT and G95L is very unlikely to be the result of differential subcellular localization or steady-state protein level. This later point justifies our focus on G95L in Figures 5 and 6.

7) Authors state that late stage inhibition of HIV-1 entry is regulated by glycine 95 (subsection “Glycine-95 is important for the antiviral functions of IFITM3”), these data are shown in a single bar chart in Figure 2F and a composite supplementary Figure 2C. In supplementary Figure 2C, the authors should explain how the quantification relates to the Western blot panels, presumably all the quantification graphs refer to the right-hand panels of the westerns labelled “virus supernatants”, but this needs to be clarified. Furthermore, could the (arguably small) differences seen for wt vs. glycine mutant in right-hand panel (supernatants) not be a consequence of the differences seen in the left-hand panel (i.e. expression levels in whole cell lysates), the latter potentially due to the effects shown for IFITM3 on HIV protein synthesis (Lee et al., 2018, “IFITM proteins inhibit HIV-1 protein synthesis”)? It is not clear to me that these data wholly support the authors interpretation that late stage inhibition of HIV-1 entry is regulated by glycine 95 and the conclusion that “oligomerization-defective G95L mutant lessened the impact of virion-associated IFITM3 on HIV-1, suggesting that IFITM3 oligomers are needed to maximally reduce virion infectivity”.

We have moved all of the data pertaining to the effect of IFITM3 on HIV virion infectivity to the main figures (now Figure 3). We also included the effect of the G91L mutation to this set of the results, such that a more general conclusion can be made about the role of the GxxxG motif in this activity. We find that G91L completely abrogated this anti-HIV activity while G95L inhibited it partially. To better understand the mechanistic reason for loss of anti-HIV function by G91L/G95L, we examined HIV Env levels in virus producing cells (Figure 3C) and virus-containing supernatants (i.e. virions) (Figure 3D). It appears that the reviewer misunderstood these data. It was previously shown by us and others (Compton et al., 2014, Tartour et al., 2014, Compton et al., 2016, Yu et al., 2015, Ahi et al., 2020) that IFITM3 impairs HIV Env processing in virus-producing cells, resulting in decreased levels of Env incorporated into virions, and IFITM3 itself incorporates into virions. However, there is also evidence that IFITM3 impairs Env function in a qualitative manner (Compton et al., 2014 and Ahi et al., 2020), and this seems to correlate with the extent of IFITM3 incorporation into virions. Here, we perform quantitative measurements of Env and IFITM3 in virions/virus supernatants. An example of one virion blot is seen in Figure 3D, and virus producing cells from the same experiment are provided in Figure 3C, to confirm that the impact of IFITM3 on Env originates in virus-producing cells (as the reviewer suggests). We then show composite data from multiple blots of virions/virus supernatants in order to glean a mechanistic understanding of why G91L and G95L result in loss of impairment of virion infectivity. We quantified levels of Env gp120 (Figure 3E), Env gp41 (Figure 3F), and IFITM3 (Figure 3G) in virions/virus supernatants in order to identify correlates of antiviral function which have been lost by G91L and G95L mutations. We observe that G91L and G95L rescue gp120 and gp41 levels in HIV-1 virions, suggesting that the GxxxG motif must be important for the reduction of Env in virions. However, G91L and G95L restore Env levels to a similar extent yet the same mutations differentially impact HIV infectivity, suggesting that Env levels do not completely explain this antiviral function. However, we observe that IFITM3 encoding G91L is less incorporated into virions relative to G95L and WT, suggesting that our data allows us to tease apart the contributions made by Env and IFITM3 in the loss of virion infectivity (Figure 3G). We now discuss this in detail in the Results: “Since the G91L and G95L mutations differentially impact the restriction of HIV-1 virion infectivity by IFITM3, it is therefore unlikely that Env quantity in virions fully accounts for this restriction. However, we observed that the G91L and G95L mutations strongly impaired the ability of IFITM3 itself to incorporate into virions (Figure 3G), and the extent of virion incorporation correlated with the measured impact on HIV-1 infectivity (Figure 3B). Together, our results demonstrate that the dual antiviral functions performed by IFITM3 (early-stage inhibition of virus entry and late-stage inhibition of virion infectivity) are critically regulated by the GxxxG motif. Interestingly, the former function depends on primarily on glycine-95, while the latter function depends more on glycine-91.”

It is unlikely that the process outlined by Lee et al., 2018 is at play here, because we are studying virion infectivity which has been normalized to levels of p24 Gag in the supernatant. That means that we can exclude the impact of IFITM3 on viral protein translation and focus solely on infectivity per ng amount of p24 Gag. It remains possible that IFITM3 WT leads to loss of Env in virus-producing cells due to a process outlined by Lee et al. However, here we are not concerned with *why* Env levels are lower in producer cells, but rather, whether differences in Env quantity account for the antiviral activity of IFITM3 WT and the lack thereof of G91L/G95L. Our data suggest that Env quantity in virions only partially explains the full antiviral potential of IFITM3 when expressed in virus-producing cells, which has been recently outlined and discussed in a recent article from our lab (Ahi et al., 2020).

8) Figure 3 (on oligomerisation) is confusing- the circular symbols are not explained and I was unclear what the 3 columns represented and whether the labels at the top (mCherry, IFITm3-YFP and IFITM3-mCherry) pertained to the columns (in which case they need to be centered over the columns) or were a legend. Accordingly, I could not interpret these data or the bar charts below the IF which, again, used these undefined circular symbols. The figure legend refers to filled red and filled yellow circles, which I could not see on the figure. I have the same problem interpreting Figure 6 (analagous experiments with PRRT2).

We apologize for the confusion. It appears that the automatic formatting performed by *eLife*’s manuscript submission portal prevented some of the colored circles from showing correctly. We have ensured that this will not happen again in the revised figure, which is now Figure 4. The labels at the top of the figure represent the legend. To make this more apparent, we drew a black box around those labels, and moved them to the bottom of the panels. We are confident that this figure, and the others containing similar data, will be entirely interpretable now.

9) The co-IP and native PAGE data lack loading controls (Figure 4) and I was also unclear why i. the heterologous pairs had not been tried for co-IPs experiments ie WT with G95L, ii. there was no replicates or quantitation for the native PAGE (i.e. an equivalent panel as for 4B).

To satisfy this request, we added additional loading controls (heavy chain for the IP fraction and actin for the whole cell lysates fraction) to the co-IP data which now appears in Figure 5.

We appreciate the reviewer’s demand that heterologous pairs (IFITM3 WT + IFITM3 G95L) be examined by co-IP. We have included these novel results into Figure 5—figure supplement 1B and C. We found that the G95L mutation introduced into one member of the pair resulted in an intermediate loss of oligomerization, mirroring our results using the FRET-based assay (Figure 4B and 4C). Furthermore, we added the quantitation of three replicates of the blue native PAGE, which now appears in Figure 5D. We thank the reviewer for pointing out these omissions.

10) Using the FliptR system, authors report that “membrane order enhancement tracks with a functionally competent form of IFITM3 but not with a loss-of-function mutant”. It would have been useful to assess one of the many other loss of function IFITM mutants that have been described in the literature, e.g. the S-palmitoylation defective mutants or an amphipathic helix-lacking mutant (especially since some of the same authors discovered IFITM3's amphipathic helix). Further, while positive and negative controls for membrane order were used (supp Figure 5), it would help strengthen the data on membrane rigidity if authors had used an additional method, especially one that has been used previously to study IFITM3 e.g. Laurdan.

We agree that testing additional IFITM3 mutants for their capacity to promote membrane order would contribution additional novel insight and strengthen the appropriateness of this manuscript for *eLife*. To that end, we introduced new data into Figure 6—figure supplement 1E. Here, we show that IFITM3 lacking a functional amphipathic helix (S61A, N64A, T65A) is incapable of increasing membrane order. Since both G95L and S61A/N64A/T65A exhibit a loss of antiviral activity (as shown in this manuscript and in Chesarino et al., 2017, respectively), this further confirms the functional link between virus restriction and elevated membrane order. However, unlike G95L, we show that S61A/N64A/T65A does not result in a loss of oligomerization (this result added to Figure 4—figure supplement 1). These novel data support our hypothesis that the antiviral functions of IFITM3 require both an amphipathic helix AND the oligomerization determinants conferred by the GxxxG motif. This significant finding is discussed in the Discussion: “Here, we show for the first time that the amphipathic helix of IFITM3 is critical for the membrane order enhancement by IFITM3. Since we also show that glycine-95 of the GxxxG motif is also required for membrane order enhancement, our data suggest that oligomerization “activates” the membrane deforming activity of the amphipathic helix, and as a result, its antiviral potential. It is possible that local insertion of multiple amphipathic helices into stretches of membrane is required for inhibition of virus fusion, and IFITM3 oligomers provide a means to fulfil that requirement.”

Lastly, as the reviewer recommended, we performed Laurdan staining as a complimentary method to study the impact of IFITM3 on membrane order. We found that the results with Laurdan mirrored our results using the FliptR approach, such that IFITM3 WT increased order while G95L did not. These new data are presented in Figure 6—figure supplement 1C and D.